# Neuro-Symbolic Decoding of Neural Activity

**Yanchen Wang**[*]
Columbia University

**Joy Hsu**[*]
Stanford University

**Ehsan Adeli**[†]
Stanford University

**Jiajun Wu**[†]
Stanford University

## Abstract

We propose NEURONA, a neuro-symbolic framework for fMRI decoding and concept grounding in neural activity. Leveraging image- and video-based fMRI question-answering datasets, NEURONA learns to decode interacting concepts from visual stimuli based on patterns of fMRI responses, integrating symbolic reasoning and compositional execution with fMRI grounding across brain regions. We demonstrate that incorporating structural priors (e.g., compositional predicate-argument dependencies between concepts) into the decoding process significantly improves both decoding accuracy over precise queries, and notably, generalization to unseen queries at test time. With NEURONA, we highlight neuro-symbolic frameworks as promising tools for understanding neural activity. Our code is available at `https://github.com/PPWangyc/neurona`.

## 1 Introduction

A long-standing hypothesis in cognitive science, the Language of Thought (LoT) hypothesis (Fodor, 1975), proposes that human cognition operates over structured representations that compose systematically. Rather than treating concepts as isolated units, many theories propose that the brain organizes knowledge into compositional structures—such as predicates and their arguments—that enable flexible inference. In this work, we study whether explicitly modeling such structure can improve neural decoding from functional magnetic resonance imaging (fMRI), with the goal of predicting high-level semantic content from patterns of brain activity (Mitchell et al., 2008). More broadly, we aim to assess whether incorporating structural priors yields decoding models that are more accurate, precise, and generalizable.

There has been extensive literature on semantic representation and concept decoding from fMRI in the past decades, with several influential works studying how concepts are organized and grounded across the cortex (Mitchell et al., 2008; Palatucci et al., 2009; Huth et al., 2016; Pereira et al., 2018). Recent advances in machine learning have further accelerated data-driven approaches to neural decoding. However, most large-scale fMRI decoding studies focus on either isolated concepts or holistic stimulus reconstruction (Nishimoto et al., 2011; Naselaris et al., 2011; Chen et al., 2023a; Takagi & Nishimoto, 2023; Scotti et al., 2023; Chen et al., 2023b), leaving open a central question: how can we decode high-level relational meaning—interactions between multiple visual concepts and the relations that bind them—from neural responses? Concretely, we ask whether decoding predicate-driven concepts (e.g., `holding`) can be improved by explicitly accounting for their constituent arguments (e.g., `person` and `baseball-bat`) and their representations across brain regions.

To explore these questions, we leverage rich data from image- and video-based fMRI datasets, which naturally encode complex and compositional semantics. Naturalistic stimuli such as images and videos often involve multiple interacting concepts (e.g., a person holding a baseball bat), making them well-suited for probing how to decode entities and their relations from neural activity. Hence, we propose challenging fMRI question answering (fMRI-QA) datasets derived from BOLD5000 (Chang et al., 2019) and CNeuroMod (Gifford et al., 2024; Boyle et al., 2023), pairing fMRI recordings with structured queries about the corresponding visual stimuli (e.g., "Is there a person holding a baseball bat?"). Our **BOLD5000-QA** and **CNeuroMod-QA** datasets evaluate decoding of fine-grained, compositional visual semantics from fMRI responses.

---

[*]Equal contribution.
[†]Equal mentorship.

We find that neither simple linear models nor purely end-to-end neural decoding models are sufficient for solving this task. Linear models lack the capacity to capture interactions between multiple interacting components Naselaris et al. (2011), while large neural decoders (e.g., those with language model backbones) tend to encode stimuli holistically, without explicitly modeling modular concepts or their relations, leading to coarse alignment between neural activity and language Takagi & Nishimoto (2023). To address these limitations, we propose a neuro-symbolic approach to fMRI-QA that integrates the compositionality of symbolic systems with the expressivity of neural networks: each query is mapped to a symbolic expression that composes concepts, and brain activity is routed through corresponding concept modules (implemented as neural networks) to produce an answer. Crucially, we incorporate various structural priors into the model's decoding process, by defining candidate entity representations in neural activity and specifying how predicate-level concepts should compose over these entity representations according to the symbolic expression. From this paradigm, we propose **NEURONA**, a NEURO-symbolic framework for decoding in Neural Activity, which integrate symbolic reasoning and compositional execution with fMRI grounding.

We evaluate NEURONA on **BOLD5000-QA** and **CNeuroMod-QA**, and demonstrate that our neuro-symbolic framework significantly outperforms baseline neural decoding methods, and importantly, exhibits strong generalization to unseen test queries. Ablation studies further highlight the functional role of compositional structure: conditioning predicate modules on the regions associated with their subject and object arguments consistently yields large performance gains in neural decoding. These priors guide the model to recover high-level semantics conditioned on constituent entity groundings, showing that relational meaning is better predicted across multiple co-activated brain regions via its arguments, rather than localized to a single region or to multiple regions without guidance.

To summarize, the main contributions of this work are:

- We introduce the **BOLD5000-QA** and **CNeuroMod-QA** datasets for fMRI question answering, pairing fMRI responses with compositional queries derived from visual stimuli to emphasize decoding of fine-grained visual semantics.

- We propose **NEURONA** as a neuro-symbolic framework for neural decoding that integrates structural priors over concept composition with grounding in neural activity.

- We show that NEURONA significantly outperforms strong neural decoding baselines and generalizes better to unseen compositional queries, and identify predicate-argument dependencies as a key driver of performance gains through targeted ablations.

## 2 RELATED WORKS

**Visual decoding from fMRI.** Reconstructing visual content from fMRI signals has become a central research focus of works in the field, with many approaches leveraging state-of-the-art generative backbones for the task, following early studies (Thirion et al., 2006; Miyawaki et al., 2008; Kay et al., 2008; Naselaris et al., 2009; Nishimoto et al., 2011). Takagi et al. demonstrated that a pre-trained diffusion model can reconstruct high-resolution images from fMRI (Takagi & Nishimoto, 2023). MinD-Vis uses masked brain modeling with a diffusion model for semantically faithful image generation (Chen et al., 2023a). MindEye projects fMRI into a CLIP embedding space and applies a diffusion prior for pixel-level synthesis (Scotti et al., 2023). Extending to video, MinD-Video and NeuroCLIP incorporate spatiotemporal masked modeling and keyframe-perception flow cues, respectively, into diffusion-based reconstruction (Chen et al., 2023b; Gong et al., 2024). These visual reconstruction works focus on recovering stimulus appearance from neural data; in contrast, rather than generating pixel-level images or videos, our work targets decoding of fine-grained semantic concepts and their relations from neural activity.

**Concept grounding.** Several influential works have focused on how semantic information is represented and organized across the cortex. As a representative work, Huth et al. used voxel-wise encoding models with natural narrative stimuli to construct a semantic atlas, showing that different semantic domains selectively ground to distinct brain regions (Huth et al., 2016). Mitchell et al. predicted fMRI patterns for concrete nouns using corpus-derived semantic features, showing generalization to unseen words (Mitchell et al., 2008). SOC enabled zero-shot decoding by mapping fMRI to semantic codes and recognizing novel object categories (Palatucci et al., 2009). Pereira et al.

introduced a general decoder that maps fMRI into a shared semantic space, enabling generalization from limited data (Pereira et al., 2018). Beyond semantic mapping, several studies have also explored how concepts are organized in the brain (Frankland & Greene, 2015; Eichenbaum, 2001). For example, abstract concept processing and rule-like representations have been linked to prefrontal and medial temporal regions across tasks (Quiroga et al., 2005; Rey et al., 2015; Tian et al., 2024; Dijksterhuis et al., 2024). Additionally, single-neuron recordings in the human prefrontal cortex have revealed neurons that encode abstract task rules independently of sensory or motor details (Mian et al., 2014). In contrast to these prior works, our approach emphasizes compositional concept grounding in the neural decoding process, with focus on functional compositionality, where we explicitly model not only individual concepts, but the relations between them. This modeling enables a targeted study of how predicate-level meaning can be decoded through structured grounding over brain regions.

**fMRI-question answering.** Recent works have explored using fMRI data for question-answering by integrating large vision-language models (VLMs). These methods typically map neural activity to visual embeddings, then generate answers using pre-trained VLMs. For example, SDRecon (Takagi & Nishimoto, 2023) projects fMRI signals into BLIP (Li et al., 2022) embeddings for captioning; BrainCap (Ferrante et al., 2023) maps fMRI to GIT (Wang et al., 2022) features for visual description; and UMBRAE (Xia et al., 2024) aligns fMRI to multimodal embeddings with subject-specific tokenization and answers questions via LLaVA (Liu et al., 2023). These methods commonly use BLEU scores (Papineni et al., 2002) to measure alignment with ground-truth text, but they do not verify whether the predicted answer captures exact underlyng concepts and relational structure. In contrast, our framework first grounds fMRI activity to modular concept representations and then performs structured reasoning over the resulting symbolic form, enabling precise, accurate, and generalizable question answering.

## 3 METHOD

### 3.1 NEURO-SYMBOLIC FRAMEWORK

We introduce NEURONA, a neuro-symbolic framework for neural decoding with concept grounding in fMRI activity. Neuro-symbolic models decompose queries into symbolic expressions over concepts, and then differentiably execute those expressions over input representations, using learned concept grounding modules to perform a variety of downstream tasks (Yi et al., 2018; Mao et al., 2019; Hsu et al., 2023b; Mao et al., 2025). In this paradigm, each concept (e.g., `person`, `holding`) is associated with a small neural network that maps entity-centric representations from input data to a predicted semantic signal, enabling intermediate grounding to be learned from weak supervision of downstream tasks. A differentiable executor composes concept module outputs according to the structure of the symbolic expression, allowing end-to-end training to predict the final answer.

In this work, we propose NEURONA as a differentiable neuro-symbolic model that adapts and extends the LEFT framework (Hsu et al., 2023a) to the fMRI decoding setting. LEFT was originally designed for concept grounding across visual domains (e.g., 2D images, 3D scenes), where a set of candidate entities (e.g., object proposals) is typically specified a priori. In contrast, our setting introduces a unique challenge: for fMRI there is no canonical notion of entities analogous to objects in a scene: the relevant neural representations for a given concept are not provided by supervision and must be inferred. Accordingly, NEURONA jointly (i) evaluates hypotheses over candidate neural entities derived from parcelled fMRI signals and (ii) learns how these entities should be composed to support precise decoding. To make differentiable program execution compatible with this uncertainty, NEURONA introduces execution semantics that replace hard selection with a soft, role-conditioned aggregation mechanism that guides predicate grounding, which we detail in below sections.

Concretely, we train NEURONA on the fMRI question answering (fMRI-QA) task (See Figure 1). Each example consists of two elements. The first is a symbolic query about the visual stimulus $v$ (e.g., "Is there a person holding a baseball bat?"). The second is an fMRI recording $f \in \mathbb{R}^{N \times T}$ measured while the stimulus is viewed, where $N$ is the number of input parcels and $T$ is the number of time points. The goal is to predict an answer $a$, either as a Boolean label (e.g., True/False) or as a classification over an answer concept vocabulary. Crucially, no supervision is provided on which brain regions are relevant to each concept.

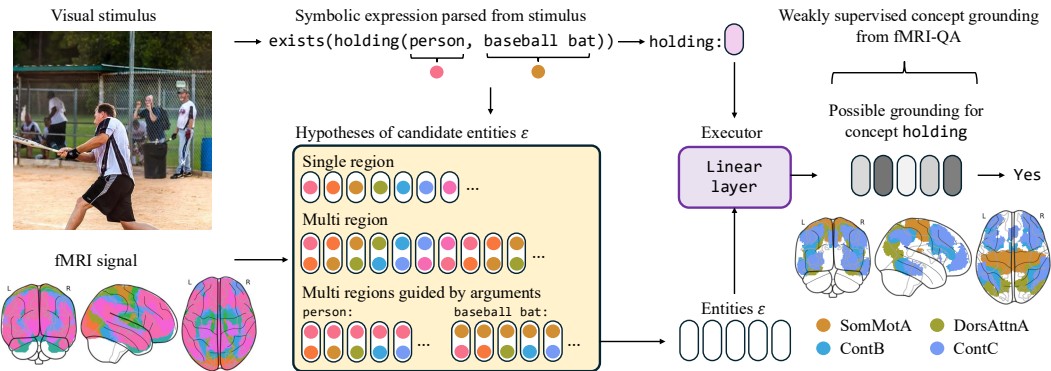

Figure 1: NEURONA is a neuro-symbolic framework for neural decoding that parses each query into a symbolic expression and maps the accompanying fMRI recording into candidate parcel-level embeddings. It grounds concepts in the expression to these candidate parcels with learned linear concept modules, optionally guided by predicate-argument structure, and composes the grounded scores to answer the question. Supervision is provided only by the final answer, which enables learning of intermediate groundings.

To define candidate neural entities, we map the fine-grained networks to $P$ functional networks using a standard functional atlas. We experiment with Yeo-7 and Yeo-17 (Yeo et al., 2011), DiFuMo-64 and DiFuMo-128 (Dadi et al., 2020), and Schaefer-100 (Schaefer et al., 2018) atlases, where $P$ is 7, 17, 64, 128, 100 respectively. All atlases yield consistently strong performance with NEURONA; in the main text we report results using Yeo-17, and provide results across atlases in Appendix A. After parcellation, we obtain $P$ parcel-level fMRI signals and encode them into a unified set $\mathcal{E}$ of parcellation embeddings $\{e_1, \ldots, e_P\}$. These parcel embeddings define candidate neural entities onto which concepts may ground.

In NEURONA, each concept is associated with a grounding module (i.e., a neural network) that maps from the candidate entity embeddings to concept-specific scores—intuitively, identifying which parcels (or parcel pairs) best support predicting that concept. A differentiable executor then composes grounded concepts according to the symbolic expression to produce a final output $a$ (a Boolean value or a distribution over answer classes). This modular structure enables NEURONA to decode compositional semantics from fMRI and to test hypotheses about how concept grounding should be guided across distributed neural signals without requiring direct supervision on intermediate groundings. In the following sections, we describe our grounding formulation, the hypotheses we test, and the training objective.

## 3.2 GROUNDING CONCEPTS ONTO NEURAL ACTIVITY

Given parcel embeddings $\mathcal{E} = [e_1, \ldots, e_P]$, where each entity is embedded to dimension $d$, concept grounding aims to assign each concept, out of $C$ total concepts, a score over these candidate entity embeddings.

**Unary concepts.** To model unary concepts (e.g., subjects and objects such as `person` and `baseball-bat`), we score each concept over entity embeddings $\mathcal{E}$ using a linear projection with weights $\mathcal{W}_{\text{unary}} \in \mathbb{R}^{d \times C}$ and bias $\mathbf{b}_{\text{unary}} \in \mathbb{R}^C$, producing concept-specific evidence over parcels. For each entity $e_p$, the predicted logits are computed via this linear layer; the overall predicted logits $\mathcal{Z}_{\text{unary}} \in \mathbb{R}^{P \times C}$ is the grounding logits of all concepts, and the grounding score for concept $c$ is $G_{\text{unary}}(c) = [z_{1c}, \ldots, z_{Pc}]^\top \in \mathbb{R}^P$, where $z_{p,c}$ denotes the logit for concept $c$ at parcel $p$.

**Relational concepts.** To model relational concepts (e.g., predicates such as `holding`), we compute logits over parcel pairs. We augment the entity embeddings $\mathcal{E}$ with learnable embeddings $\mathcal{E}_b \in \mathbb{R}^{P \times d}$ and form $\mathcal{E}_c \in \mathbb{R}^{P \times 2d}$, where $\mathcal{E}_c = \mathcal{E} \oplus \mathcal{E}_b$ and $\oplus$ denotes concatenation. Here, $\mathcal{E}_b$ provides features that represent each parcel. For each pair of parcels $(i, j)$, we concatenate their embeddings $e_{c_i} \oplus e_{c_j}$

and apply a learnable transformation $\mathcal{W}_{\text{pair}} \in \mathbb{R}^{4d \times d}$ to obtain a pairwise representation $\mathcal{E}'_{ij} \in \mathbb{R}^d$. We then apply a linear classifier to compute the logits for each pair, yielding $\mathcal{Z}_{\text{binary}} \in \mathbb{R}^{P \times P \times C}$, which represents the grounding logits of all concepts. The grounding score for a concept $c$ is $G_{\text{binary}}(c) = [z_{ij,c}]_{1 \leq i,j \leq P} \in \mathbb{R}^{P \times P}$.

**From groundings to answers.** The grounded scores are used (and optionally composed) to answer fMRI-QA queries. Let $c_s$, $c_o$, and $c_p$ denote the subject, object, and predicate concepts in a query. For Boolean queries, we optionally condition the predicate grounding $G_{\text{binary}}(c_p)$ on the unary groundings $G_{\text{unary}}(c_s)$ and $G_{\text{unary}}(c_o)$, and aggregate the resulting scores. Then, we apply a sigmoid over the scores, and threshold the result to produce a binary decision in inference.

For concept classification queries with answer vocabulary $\mathcal{V}$, we compute unary and relational logits for each candidate answer $v \in \mathcal{V}$ from the precomputed tensors, treating them as the unary similarity $\mathcal{S}_{\text{unary}}$ and relational similarity $\mathcal{S}_{\text{binary}}$ scores as follows

$$S_{\text{unary}} = [z_{iv}]_{1 \leq i \leq P, v \in \mathcal{V}} \in \mathbb{R}^{P \times |\mathcal{V}|}, \quad S_{\text{binary}} = [z_{ij,v}]_{1 \leq i,j \leq P, v \in \mathcal{V}} \in \mathbb{R}^{P \times P \times |\mathcal{V}|}. \tag{1}$$

When subject and object groundings are available, we compute guided scores

$$S_{\text{unary}}^{\text{guided}} = G_{\text{unary}}(c_s)^\top S_{\text{unary}} + G_{\text{unary}}(c_o)^\top S_{\text{unary}} \in \mathbb{R}^{|\mathcal{V}|}, \tag{2}$$

$$S_{\text{binary}}^{\text{guided}} = G_{\text{unary}}(c_s)^\top \left( G_{\text{unary}}(c_o)^\top S_{\text{binary}} \right) \in \mathbb{R}^{|\mathcal{V}|}, \tag{3}$$

and combine them as $S^{\text{final}} = S_{\text{unary}}^{\text{guided}} + S_{\text{binary}}^{\text{guided}} \in \mathbb{R}^{|\mathcal{V}|}$, with the final predicted concept selected by $\hat{v} = \arg\max_{v \in \mathcal{V}} S_v^{\text{final}}$. This yields a unified, differentiable formulation for grounding unary and relational concepts for both Boolean and vocabulary-based queries.

## 3.3 Testing hypotheses of grounding structures

We evaluate five hypotheses about how concepts may be grounded over parcels and parcel pairs. Because NEURONA is modular, we can explicitly test guided grounding by conditioning predicate representations on their arguments rather than treating each concept independently. Since relational groundings are defined over parcel pairs, we compare hypotheses in per-parcel space by reducing pairwise scores via aggregation $\tilde{G}_{\text{binary}}(c)_i = \frac{1}{P} \sum_{j=1}^{P} [G_{\text{binary}}(c)]_{ij}$.

**H1: Single-region localized.** Concepts are best supported by a single region, where the grounding score is defined as $G_{\text{H1}}(c) = G_{\text{unary}}(c)$.

**H2: Multi-region co-activation.** Concepts are best supported by co-activation across region pairs, where the grounding score is defined as $G_{\text{H2}}(c_p) = \tilde{G}_{\text{binary}}(c_p)$.

**H3: Subject-guided multi-region.** Predicate grounding is guided by subject evidence, with the grounding score defined as $G_{\text{H3}}(c_p) = \tilde{G}_{\text{binary}}(c_p) + G_{\text{unary}}(c_s)$.

**H4: Object-guided multi-region.** Predicate grounding is guided by object evidence, with the grounding score defined as $G_{\text{H4}}(c_p) = \tilde{G}_{\text{binary}}(c_p) + G_{\text{unary}}(c_o)$.

**H5: Full argument-guided multi-region.** Predicate grounding is guided by both arguments, with the grounding score defined as $G_{\text{H5}}(c_p) = \tilde{G}_{\text{binary}}(c_p) + G_{\text{unary}}(c_s) + G_{\text{unary}}(c_o)$.

Together, these hypotheses define different ways of structuring grounding over $\mathcal{E}$, allowing us to test whether structure-guided grounding improves decoding. Full derivations and experimental details are provided in Appendix G.

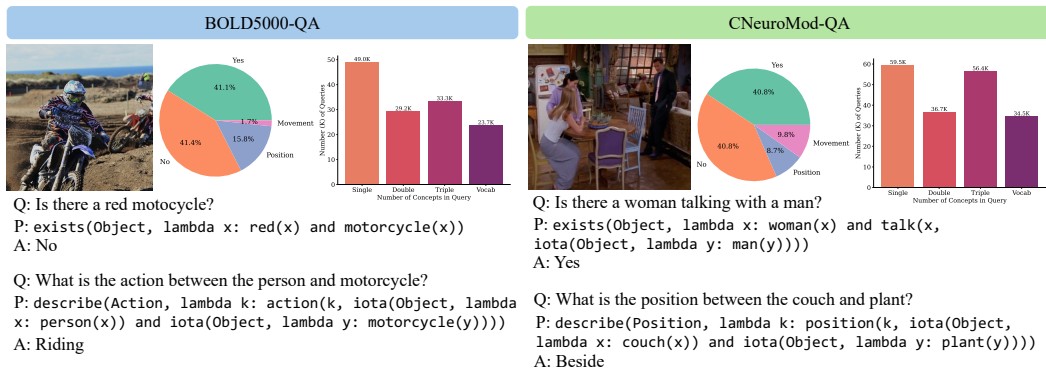

Figure 2: We include example queries and dataset distribution overviews for BOLD5000-QA and CNeuroMod-QA; both datasets span diverse queries and tasks.

### 3.4 TRAINING OBJECTIVE

We train NEURONA in the fMRI-QA setting, where ground-truth answers are provided for each query, but no supervision is provided on intermediate concept groundings, as none are available. We optimize a standard cross-entropy objective over the model's predicted answer distribution $\mathcal{L}_{\text{CE}} = -\sum_{k=1}^{K} a_k \log(\hat{a}_k)$, where $\hat{a} \in \mathbb{R}^K$ is the predicted probability distribution over $K$ answer classes after executing the full symbolic expression, and $a$ is the ground-truth label.

## 4 DATASETS

To train and evaluate NEURONA, we construct fMRI question answering (fMRI-QA) datasets by adapting existing large-scale fMRI-vision datasets. Our goal is to derive structured, compositional supervision from rich visual stimuli. To do so, for each stimuli, we use a pre-trained vision-language model to produce a scene graph that captures both object-level and relational semantics, for example, unary entities such as `person`, `baseball-bat`, and higher-arity predicates describing their interaction such as `holding (person, baseball-bat)`.

We then convert the scene graph into a set of structured question-answer pairs, where each question corresponds to a symbolic query, and each answer provides supervision for neural decoding. For example, given the above scene graph, we construct questions such as: "What is the relation between person and baseball-bat?". We additionally generate negative samples by sampling concepts from other stimuli in the dataset. This process yields diverse fMRI-QA examples covering both unary entities and higher-arity relations, with precise answers. We apply this procedure to BOLD5000 (Chang et al., 2019) and CNeuroMod (Gifford et al., 2024; Boyle et al., 2023) (See Figure 2), to create the BOLD5000-QA and CNeuroMod-QA datasets. More dataset details can be found in Appendix F.

**BOLD5000-QA.** The BOLD5000 dataset is a large-scale fMRI dataset collected while participants viewed naturalistic images. The dataset consists of approximately $5,000$ distinct images from three datasets: Scene Images (Xiao et al., 2010), COCO (Lin et al., 2014), and ImageNet (Deng et al., 2009). Four participants viewed these images while undergoing whole-brain fMRI scanning. To create BOLD5000-QA, we follow the process above, and generate queries containing $4,258$ unary and $135$ relational concepts, with $133,146$ train and $2,095$ test examples.

**CNeuroMod-QA.** The CNeuroMod dataset is a large-scale fMRI dataset that includes recordings of participants watching full-length naturalistic videos. Specifically, we build upon the Friends dataset, and set season $1$ to $5$ to be the train samples, and unseen season $6$ to be the test. To create CNeuroMod-QA, we sample video frames based on motion energy, defined as the absolute difference between consecutive frames. We then extract scene graphs for each sampled frame, aggregate the changes, and construct corresponding symbolic queries. The CNeuroMod-QA dataset includes $1,966$ unary and $106$ relational concepts, with $157,046$ train and $30,059$ test examples.

Table 1: We evaluate NEURONA on BOLD5000-QA and CNeuroMod-QA, comparing its performance to prior fMRI language decoding models and a linear baseline.

| | BOLD5000-QA | | | | CNeuroMod-QA | | | |
|---|---|---|---|---|---|---|---|---|
| Method | Overall | Action | Position | T/F | Overall | Action | Position | T/F |
| Linear | 0.4692 | 0.2069 | 0.1778 | 0.5260 | 0.4638 | 0.3043 | 0.1285 | 0.5192 |
| UMBRAE | 0.4754 | 0.2069 | 0.1746 | 0.5340 | 0.4642 | 0.1432 | 0.1762 | 0.5378 |
| SDRecon | 0.4711 | 0.2414 | 0.1937 | 0.5248 | 0.4430 | 0.1350 | 0.1481 | 0.5238 |
| BrainCap | 0.4773 | 0.1937 | 0.1724 | 0.5551 | 0.4417 | 0.1257 | 0.1477 | 0.5112 |
| NEURONA (Ours) | **0.7041** | **0.6207** | **0.5079** | **0.7407** | **0.7046** | **0.6514** | **0.5746** | **0.7250** |

Table 2: Generalization results of NEURONA and prior work on unseen queries.

| | BOLD5000-QA | | | | CNeuroMod-QA | | | |
|---|---|---|---|---|---|---|---|---|
| Method | Overall | Action | Position | T/F | Overall | Action | Position | T/F |
| Linear | 0.4587 | 0.0690 | 0.0794 | 0.5231 | 0.4143 | 0.0398 | 0.0323 | 0.5003 |
| UMBRAE | 0.4501 | 0.1379 | 0.0984 | 0.5191 | 0.4502 | 0.1205 | 0.1384 | 0.5225 |
| SDRecon | 0.4702 | 0.2414 | 0.1937 | 0.5237 | 0.4341 | 0.1236 | 0.1473 | 0.5008 |
| BrainCap | 0.4754 | 0.1724 | 0.1937 | 0.5311 | 0.4347 | 0.1198 | 0.1402 | 0.5042 |
| NEURONA (Ours) | **0.6840** | **0.6207** | **0.4984** | **0.7184** | **0.6583** | **0.4365** | **0.5261** | **0.6991** |

## 5 RESULTS

We evaluate NEURONA on neural decoding accuracy and on the consistency of its intermediate concept groundings in the below sections. We present quantitative results in Section 5.1, qualitative analyses in Section 5.2, and a discussion of limitations and implications in Section 5.3.

### 5.1 QUANTITATIVE PERFORMANCE

We measure NEURONA's performance on both the BOLD5000-QA and CNeuroMod-QA datasets, with full predicate-argument guidance as the underlying grounding structure. We compare against baseline fMRI decoding methods, test generalization to unseen compositional queries, conduct ablations that correspond to our grounding hypotheses, and report a quantitative consistency metric for concept grounding. Additional results across atlases, across subjects, across tasks, and detailed analyses are provided in the Appendix.

**Comparison to prior works.** We first evaluate NEURONA on the fMRI question answering (fMRI-QA) task, compared against existing decoding methods: a linear baseline, UMBRAE (Xia et al., 2024), SDRecon (Takagi & Nishimoto, 2023), and BrainCap (Ferrante et al., 2023). All models are trained on subject CSI1 (BOLD5000) and subject sub-01 (CNeuroMod). As shown in Table 1, NEURONA consistently outperforms prior works across both the BOLD5000-QA and CNeuroMod-QA datasets. Relative to the strongest prior method, NEURONA achieves a 47% relative improvement. These results suggest that linear methods and purely end-to-end decoding pipelines both struggle with the fine-grained concept and relation decoding required by fMRI-QA. In particular, NEURONA demonstrates significant gains on queries about actions and positions, which involve precise relational reasoning over subject and object roles. This highlights the strength of our neuro-symbolic framework, which explicitly grounds neural activity using predicate-argument structure to answer relational queries.

**Generalization to unseen compositions.** We further measure NEURONA's ability to generalize to unseen compositions of concepts at test time, which robustly evaluates whether learned groundings support systematic recombination. Specifically, we construct evaluation splits where all training and testing queries are disjoint, with no overlapping combinations of entities and relations, forcing models to generalize compositionally. For example, the training set may contain only queries such as `in_front_of(person,surfboard)`, while the test set includes held-out compositions such as `in_front_of(surfboard,person)`. As shown in Table 12, NEURONA achieves the strongest

Table 3: We ablate our hypotheses about the structure of concept grounding on BOLD5000-QA.

| Method | BOLD5000-QA | | | |
| --- | --- | --- | --- | --- |
| | Overall | Action | Position | T/F |
| Single region | $0.6451 \pm 0.0161$ | $0.2973 \pm 0.0361$ | $0.2005 \pm 0.0047$ | $0.7293 \pm 0.0191$ |
| Multi-region (MR) | $0.6476 \pm 0.0048$ | $0.2973 \pm 0.0361$ | $0.2005 \pm 0.0047$ | $0.7324 \pm 0.0056$ |
| Subject-guided MR | $0.6678 \pm 0.0029$ | $0.2881 \pm 0.0299$ | $0.3469 \pm 0.0134$ | $0.7308 \pm 0.0024$ |
| Object-guided MR | $0.6733 \pm 0.0051$ | $0.3892 \pm 0.0752$ | $0.3429 \pm 0.0164$ | $0.7363 \pm 0.0045$ |
| Full argument-guided MR | $\mathbf{0.7102 \pm 0.0053}$ | $\mathbf{0.5965 \pm 0.0322}$ | $\mathbf{0.5378 \pm 0.0135}$ | $\mathbf{0.7425 \pm 0.0057}$ |

Table 4: We evaluate our hypotheses about structural priors on CNeuroMod-QA.

| Method | CNeuroMod-QA | | | |
| --- | --- | --- | --- | --- |
| | Overall | Action | Position | T/F |
| Single region | $0.6429 \pm 0.0013$ | $0.2165 \pm 0.0193$ | $0.2445 \pm 0.0009$ | $0.7400 \pm 0.0016$ |
| Multi-region (MR) | $0.6162 \pm 0.0027$ | $0.2165 \pm 0.0193$ | $0.2445 \pm 0.0009$ | $0.7042 \pm 0.0023$ |
| Subject-guided MR | $0.6265 \pm 0.0042$ | $0.2339 \pm 0.0043$ | $0.3722 \pm 0.0045$ | $0.7008 \pm 0.0051$ |
| Object-guided MR | $0.6872 \pm 0.0040$ | $0.6320 \pm 0.0019$ | $0.4933 \pm 0.0172$ | $0.7149 \pm 0.0045$ |
| Full argument-guided MR | $\mathbf{0.7189 \pm 0.0009}$ | $\mathbf{0.6422 \pm 0.0072}$ | $\mathbf{0.5931 \pm 0.0084}$ | $\mathbf{0.7417 \pm 0.0005}$ |

performance across both datasets, significantly outperforming all baselines. Notably, prior methods show significant performance degradation, often falling to near-chance levels when generalizing to unseen compositions. Language model-based baselines retain slightly better performance due to their use of pre-trained language models, which carry general linguistic priors. However, since they lack explicit concept grounding, their performance remains well below NEURONA. Overall, these results suggest that NEURONA learns meaningful groundings that support robust generalization to novel decoding queries.

**Ablations and hypotheses testing.** We conduct ablations corresponding to our grounding hypotheses, and evaluate how they affect decoding performance. We summarize results in Table 3 and Table 4: we compare single-region grounding, multi-region grounding without guidance, and guided grounding based on subject and/or object arguments. We report mean and standard deviation across 4 subjects in BOLD5000-QA and 3 subjects in CNeuroMod-QA. With NEURONA, we answer the following questions.

DOES MULTI-REGION GROUNDING IMPROVE PERFORMANCE? We first test whether allowing concepts to ground to combinations of regions improves decoding performance relative to single-region grounding. As shown in Table 3 and Table 4, we observe that multi-region grounding alone does not yield consistent gains over single-region grounding in overall accuracy. This is especially evident in vocabulary classification tasks, where both variants tend to overfit to frequent labels rather than capturing fine-grained distinctions.

DOES GUIDED GROUNDING IMPROVE PERFORMANCE? Next, we evaluate whether guiding multi-region grounding using the subject and object arguments of relational concepts improves decoding. As shown, guided variants consistently outperform unguided multi-region baselines. We find that object-guided grounding is generally more effective than subject-guided grounding, and that using both arguments yields the best performance, notably on action and position queries that require precise relational reasoning. These results suggest that while multi-region grounding expands the representational space, explicit structural guidance based on predicate-argument dependencies is crucial for accurate and generalizable decoding. The consistent results on both natural image and video datasets validate NEURONA ability to conduct complex neural decoding with structural priors.

**Concept grounding consistency.** As there is no ground truth for intermediate groundings, we introduce a consistency metric that measures whether the same concept tends to ground to similar regions across different fMRI-QA instances. We calculate consistency as follows. Let a concept $c$ appear in $N$ QA examples, and let the predicted grounding in the $i$-th example be a set of regions $B^{(i)}(c) \subseteq \{1, \ldots, P\}$, where $P$ is the total number of regions, and each $B^{(i)}(c)$ is the set of regions selected as the grounding for concept $c$ in that example. We compute the frequency count for each

region $r$ as

$$\text{Count}(r) = \sum_{i=1}^{N} \mathbf{1}[r \in B^{(i)}(c)]. \tag{4}$$

Then, the score for concept $c$ is defined as

$$\text{Consistency}(c) = \frac{1}{|R|} \sum_{r \in R} \frac{\text{Count}(r)}{N}, \tag{5}$$

where $R$ is the set of all regions that appear in any grounding of $c$, and $\frac{\text{Count}(r)}{N}$ is the fraction of times region $r$ was selected. Our proposed metric captures how concentrated a concept's groundings are: a score of 1.0 indicates that all instances select the same region set, whereas lower scores indicate greater variability.

We report consistency scores over all concepts in BOLD5000-QA and CNeuroMod-QA. As a baseline, we define a null model that randomly assigns region groundings using a multinomial distribution, preserving the exact total sample count for each concept while distributing sam-

Table 5: Consistency of concept grounding between NEURONA and a multinomial null model.

|  | BOLD5000-QA | CNeuroMod-QA |
|---|---|---|
| Null | $0.6267 \pm 0.0067$ | $0.6424 \pm 0.0057$ |
| NEURONA | $\mathbf{0.8475 \pm 0.0186}$ | $\mathbf{0.8592 \pm 0.0084}$ |

ples uniformly across all regions. This ensures that the null model retains the same structure as the observed data (i.e., same total grounding counts per concept) while randomizing only the region assignments. We run this null model 10 times and report the mean and standard deviation in Table 5. We see that NEURONA achieves significantly higher consistency than the null baseline, suggesting that its learned groundings are reproducible across stimuli, rather than arising from chance.

## 5.2 QUALITATIVE ANALYSES

Finally, we qualitatively examine how NEURONA decodes relational concepts via intermediate groundings, focusing on how predicate groundings vary with different subject-object pairs. Figure 3 shows examples from both BOLD5000-QA and CNeuroMod-QA, where we project grounding scores onto network parcellations that define candidate entities. We observe that the same predicate (e.g., `hold` or `look`), can be best decoded from different regions depending on the object. For example, in BOLD5000, `holding (person, kite)` is best decoded using the Control B network, while `holding (person, surfboard)` relies on both the Somatomotor B and Control A networks. This pattern suggests that predicate decoding benefits from argument-dependent modulation, consistent with our quantitative results on guided grounding.

Interestingly, the model-inferred groundings are not confined to early visual areas. For example, objects such as `baseball-bat` and `surfboard` often receive high grounding scores in somatomotor-associated networks. This qualitative pattern is broadly consistent with prior work linking perception of action-related objects with engagement of motor and premotor systems (Martin, 2007; Gallese et al., 1996). In addition, both `hold` and `look` are frequently decoded using prefrontal networks (e.g., the dorsal attention and salience/ventral attention networks), which have been associated with high-level cognitive control and abstract rule processing (Miller & Cohen, 2001; Quiroga et al., 2005; Tian et al., 2024). However, we emphasize that these observations reflect model-dependent decoding patterns rather than direct estimates of underlying encoding representations. Additional visualizations are provided in Appendix E.

## 5.3 DISCUSSION

Our findings show that incorporating compositional structure into the neural decoding process can significantly improve fMRI-QA performance. While NEURONA does not establish representational compositionality in neural activity, the gains from modeling predicate–argument structure as an inductive bias provide evidence that compositional assumptions can inform neural decoding. At the same time, our analyses highlight a limitation of current neuroimaging resources. We lack the combinatorial data necessary to better validate representational compositionality. Existing visual fMRI datasets Allen et al. (2022); Chang et al. (2019); Horikawa & Kamitani (2017) provide rich naturalistic stimuli (the "whole"), but rarely include the systematic, combinatorial manipulations needed to directly test how isolated "parts" compose. In addition, participants in our datasets engaged

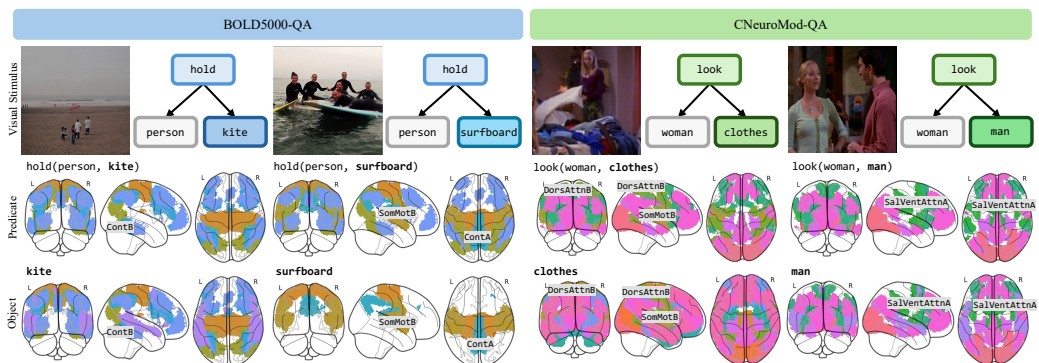

Figure 3: We show examples of learned grounding from NEURONA. On both BOLD5000-QA and CNeuroMod-QA, we see that predicate concepts ground to regions that their constituent objects arguments are grounded to, following hierarchical predicate-argument structure.

only in passive viewing, and symbolic expressions were derived through automated scene-graph parsing rather than participant-driven reasoning, limiting the cognitive conclusions that can be drawn. Moreover, we restrict candidate regions to predefined parcellations from established atlases (e.g., Yeo-7, Yeo-17, DiFuMo-64, DiFuMo-128, and Schaefer-100), which are widely used but necessarily coarse. In the Appendix, we report performance on BOLD5000-QA and CNeuroMod-QA across atlases, effect sizes between atlases, and concept grounding consistency across atlases. These results demonstrate robust decoding across different levels of spatial granularity, suggesting that our results are not tied to specific parcellation choices. However, we believe that scaling NEURONA to incorporate whole-brain voxel-level data is a promising next step.

## 6 CONCLUSION

We propose NEURONA, a neuro-symbolic framework for neural decoding and concept grounding in fMRI activity. By leveraging symbolic reasoning and compositional execution with fMRI grounding, NEURONA supports precise decoding and significantly improved generalization. Experiments on BOLD5000-QA and CNeuroMod-QA demonstrate that NEURONA outperforms strong baselines and generalizes to held-out queries, with argument-guided grounding yielding the largest gains. These results suggest that incorporating predicate-argument dependencies as a structural prior can improve relational decoding, and highlight neuro-symbolic modeling as a promising approach for for building and analyzing fMRI decoding models.

### REPRODUCIBILITY STATEMENT

We release our code at `https://github.com/PPWangyc/neurona`.

### ACKNOWLEDGMENTS

This work is in part supported by the Stanford Institute for Human-Centered AI (HAI), NIH R01AG089169, AFOSR YIP FA9550-23-1-0127, ONR N00014-23-1-2355, ONR YIP N00014-24-1-2117, ONR MURI N00014-24-1-2748, and NSF RI #2211258. JH is also supported by the Knight-Hennessy Fellowship and the NSF Graduate Research Fellowship.

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

# Supplementary for:
# Neuro-Symbolic Decoding of Neural Activity

The appendix is organized into eight main sections. Appendix A includes experiments results across atlases: performance, effect sizes, concept grounding consistency. Appendix B includes experiments results across subjects: cross-subject decoding and cross-subject consistency. Appendix C includes additional experiments results for cross-dataset transfer and fMRI retrieval tasks. Appendix D includes in-depth analyses of network ablations, statistical tests across hypotheses, fine-grained generalization, predicate argument binding, and detailed concept accuracy. Appendix E provides additional visualizations of NEURONA's grounding results. Appendix F presents more examples illustrating our fMRI-QA datasets and detail the data generation process. Appendix G describes the training procedure of NEURONA, the implementation of baseline methods, and the setup of our hypothesis ablation experiments. Appendix H details our ethics statement. Here, we also note that we use large language models to make minor improvements to writing.

## A  EXPERIMENTS ACROSS ATLASES

### A.1  PERFORMANCE ACROSS ATLASES

We report performance of NEURONA across atlases to show robustness of decoding. We map parcellated fMRI signals (1024 regions for BOLD5000, 1000 for CNeuroMod) to multiple atlases, including Yeo-7 and Yeo-17 (Yeo et al., 2011), DiFuMo-64 and DiFuMo-128 (Dadi et al., 2020), and Schaefer-100 (Schaefer et al., 2018), then train NEURONA on these candidate entities. Results in Table 6 and Table 7 show that NEURONA consistently learns across these atlases, and still significantly outperforms prior works in decoding accuracy. Yeo-17 yields the highest accuracy among all tested atlases, followed by DiFuMo-128 and Schaefer-100.

Table 6: NEURONA's performance with coarse and fine-grained atlases on BOLD5000-QA.

| BOLD5000-QA | Overall | Action | Position | T/F |
|---|---|---|---|---|
| Yeo-7 | 0.6864 | 0.5517 | 0.4730 | 0.7270 |
| Yeo-17 | 0.7041 | 0.6207 | 0.5079 | 0.7407 |
| DiFuMo-64 | 0.6992 | 0.5517 | 0.5524 | 0.7282 |
| DiFuMo-128 | 0.7026 | 0.5862 | 0.5460 | 0.7327 |

Table 7: NEURONA's performance with coarse and fine-grained atlases on CNeuroMod-QA.

| CNeuroMod-QA | Overall | Action | Position | T/F |
|---|---|---|---|---|
| Yeo-7 | 0.6969 | 0.6459 | 0.5577 | 0.7180 |
| Yeo-17 | 0.7046 | 0.6514 | 0.5746 | 0.7250 |
| Schaefer-100 | 0.7043 | 0.6549 | 0.5614 | 0.7258 |

### A.2  EFFECT SIZES BETWEEN ATLASES

To additionally evaluate the robustness of NEURONA to different parcellations, we compute Cohen's d effect sizes between QA predictions from different atlases. For each atlas pair, we compute paired effect sizes using QA predictions across the test set. As seen in Table 8 and Table 9, effect sizes are consistently small, showing that NEURONA is robust to the choice of atlas and performs reliably across a range of parcellations.

### A.3  CONCEPT GROUNDING CONSISTENCY ACROSS ATLASES

In Table 10 and Table 11, we report consistency scores averaged over all concepts in BOLD5000 and CNeuroMod, under multiple atlas configurations: Yeo-7, Yeo-17, DiFuMo-64, DiFuMo-128,

Table 8: Effect sizes between atlases in BOLD5000-QA.

| BOLD5000-QA | Yeo-7 | Yeo-17 | Difumo-64 | Difumo-128 |
|---|---|---|---|---|
| Yeo-7 | - | -0.017 | -0.133 | -0.012 |
| Yeo-17 | 0.017 | - | -0.116 | 0.006 |
| Difumo-64 | 0.133 | 0.116 | - | 0.121 |
| Difumo-128 | 0.012 | -0.006 | -0.121 | - |

Table 9: Effect sizes between atlases in CNeuroMod-QA.

| CNeuroMod-QA | Yeo-7 | Yeo-17 | Schaefer-100 |
|---|---|---|---|
| Yeo-7 | - | 0.094 | 0.089 |
| Yeo-17 | -0.094 | - | -0.006 |
| Schaefer-100 | -0.089 | 0.006 | - |

Table 10: Concept grounding consistency in BOLD5000-QA.

| BOLD5000-QA | Overall | Unary Concept | Relational Concept |
|---|---|---|---|
| Yeo-7 Null | 0.5738 | – | – |
| Yeo-7 NEURONA | 0.8207 | 0.8283 | 0.7343 |
| Yeo-17 Null | 0.5357 | – | – |
| Yeo-17 NEURONA | 0.8220 | 0.8351 | 0.6646 |
| DiFuMo-64 Null | 0.5075 | – | – |
| DiFuMo-64 NEURONA | 0.8462 | 0.8644 | 0.6064 |
| DiFuMo-128 Null | 0.5039 | – | – |
| DiFuMo-128 NEURONA | 0.8224 | 0.8380 | 0.6241 |

Table 11: Concept grounding consistency in CNeuroMod-QA.

| CNeuroMod-QA | Overall | Unary Concept | Relational Concept |
|---|---|---|---|
| Yeo-7 Null | 0.5740 | – | – |
| Yeo-7 NEURONA | 0.8437 | 0.8564 | 0.7563 |
| Yeo-17 Null | 0.5358 | – | – |
| Yeo-17 NEURONA | 0.8700 | 0.8967 | 0.6812 |
| Schaefer-100 Null | 0.5029 | – | – |
| Schaefer-100 NEURONA | 0.8346 | 0.8695 | 0.5838 |

and Schaefer-100. NEURONA achieves consistently high grounding consistency across all atlases, significantly above the null baseline. Unary concepts show higher consistency than relational ones, as expected due to their simpler structure. Across the atlases, all show consistent results, with DiFuMo-64 best for BOLD5000 and Yeo-17 best for CNeuroMod. NEURONA's concept grounding is reproducible way across different stimuli and parcellations.

Table 12: Results across subjects for BOLD5000-QA and CNeuroMod-QA.

| BOLD5000-QA | Overall | Action | Position | T/F |
|---|---|---|---|---|
| Linear | 0.4702 ± 0.0073 | 0.0677 ± 0.0815 | 0.1817 ± 0.0289 | 0.5280 ± 0.0068 |
| UMBRAE | 0.4948 ± 0.0098 | 0.2488 ± 0.0572 | 0.2092 ± 0.0401 | 0.5401 ± 0.0097 |
| SDRecon | 0.4751 ± 0.0064 | 0.2887 ± 0.0499 | 0.2005 ± 0.0047 | 0.5266 ± 0.0075 |
| BrainCap | 0.4842 ± 0.0052 | 0.2412 ± 0.0470 | 0.2005 ± 0.0047 | 0.5383 ± 0.0061 |
| NEURONA | **0.7102 ± 0.0053** | **0.5965 ± 0.0322** | **0.5378 ± 0.0135** | **0.7425 ± 0.0057** |

| CNeuroMod-QA | Overall | Action | Position | T/F |
|---|---|---|---|---|
| Linear | 0.4579 ± 0.0142 | 0.2078 ± 0.0400 | 0.1680 ± 0.0338 | 0.5184 ± 0.0153 |
| UMBRAE | 0.4588 ± 0.0425 | 0.1588 ± 0.0102 | 0.1705 ± 0.0220 | 0.5225 ± 0.0248 |
| SDRecon | 0.4368 ± 0.0012 | 0.1428 ± 0.0010 | 0.1554 ± 0.0010 | 0.5012 ± 0.0024 |
| BrainCap | 0.4422 ± 0.0026 | 0.1245 ± 0.0099 | 0.1497 ± 0.0074 | 0.5110 ± 0.0024 |
| NEURONA | **0.7189 ± 0.0009** | **0.6422 ± 0.0072** | **0.5931 ± 0.0084** | **0.7417 ± 0.0005** |

Table 13: Generalization results across subjects for BOLD5000-QA and CNeuroMod-QA.

| BOLD5000-QA | Overall | Action | Position | T/F |
|---|---|---|---|---|
| Linear | 0.4595 ± 0.0093 | 0.0308 ± 0.0312 | 0.1291 ± 0.0545 | 0.5250 ± 0.0059 |
| UMBRAE | 0.4886 ± 0.0068 | 0.1214 ± 0.0217 | 0.0914 ± 0.0212 | 0.5392 ± 0.0080 |
| SDRecon | 0.4752 ± 0.0064 | 0.2973 ± 0.0361 | 0.2005 ± 0.0047 | 0.5266 ± 0.0075 |
| BrainCap | 0.4838 ± 0.0055 | 0.2412 ± 0.0470 | 0.1996 ± 0.0040 | 0.5380 ± 0.0064 |
| NEURONA | **0.6812 ± 0.0055** | **0.4952 ± 0.0682** | **0.4696 ± 0.0190** | **0.7217 ± 0.0057** |

| CNeuroMod-QA | Overall | Action | Position | T/F |
|---|---|---|---|---|
| Linear | 0.4206 ± 0.0129 | 0.0462 ± 0.0654 | 0.1045 ± 0.0667 | 0.4986 ± 0.0052 |
| UMBRAE | 0.4487 ± 0.0040 | 0.1107 ± 0.0156 | 0.1386 ± 0.0047 | 0.5247 ± 0.0035 |
| SDRecon | 0.4365 ± 0.0014 | 0.1407 ± 0.0008 | 0.1560 ± 0.0012 | 0.5012 ± 0.0019 |
| BrainCap | 0.4382 ± 0.0012 | 0.1217 ± 0.0070 | 0.1455 ± 0.0060 | 0.5069 ± 0.0003 |
| NEURONA | **0.6676 ± 0.0119** | **0.2916 ± 0.0878** | **0.5377 ± 0.0215** | **0.7260 ± 0.0072** |

Table 14: Cross-subject consistency of concept grounding.

| | BOLD5000-QA | CNeuroMod-QA |
|---|---|---|
| Null (random) | 0.5276 | 0.5267 |
| Null (multinomial) | 0.6045 | 0.6481 |
| NEURONA | **0.7000** | **0.7027** |

# B EXPERIMENTS ACROSS SUBJECTS

## B.1 CROSS-SUBJECT DECODING

We include evaluation over all subjects on both BOLD5000-QA (4 subjects) and CNeuroMod-QA (3 subjects). For each dataset, we train an individual model for each subject and report the mean ± standard deviation across subjects. We evaluate neural decoding performance in Table 12 and our generalization split in Table 13. We see that NEURONA continues to significantly outperform all prior works.

## B.2 CROSS-SUBJECT CONSISTENCY

In addition, we evaluate cross-subject consistency in concept groundings. To evaluate whether concepts are grounded similarly across individuals, we aggregate grounding scores for each concept across all subjects and queries (for concept $c$ appearing $N$ times per subject, we obtain $N \times S$ samples where $S$ is the number of subjects). We then compute a cross-subject consistency metric that measures the similarity of the spatial grounding patterns for each concept across individuals. We evaluate NEURONA against the multinomial null model and an additional null model that randomly assigns each concept to a region subset via uniform sampling that preserves each concept's distribution across regions. Our results in Table 14 show that NEURONA's cross-subject grounding consistency scores, averaged across all concepts, are higher than both null models ($p < 0.001$), demonstrating that NEURONA's concept groundings converge to more similar sets of regions across participants.

# C ADDITIONAL TASKS

## C.1 CROSS-DATASET TRANSFER EXPERIMENTS

We report cross-dataset generalization performance by training our model on BOLD5000-QA and evaluating it on the CNeuroMod-QA test set. Since BOLD5000 spans a broader concept space, we selected overlapping queries across datasets. In the CNeuroMod test set, this includes $1,169$ queries for the action task, $2,600$ for the position task, and $23,038$ for the T/F task (out of full test set sizes of $2,912$, $2,661$, and $24,486$, respectively).

In Table 15, we compare NEURONA to UMBRAE (Xia et al., 2024), the top performing baseline model. NEURONA significantly outperforms UMBRAE across all queries, demonstrating stronger cross-dataset robustness and generalization. Notably, while overall performance of NEURONA drops, largely due to a performance gap on T/F queries, accuracy on action and position tasks remains high, indicating some degree of cross-dataset transfer. This drop is expected, as our model is trained as a subject-specific model and there is substantial variance across subjects. Additionally, the two datasets differ in preprocessing pipelines: BOLD5000 uses the DiFuMo-1024 parcellation, while CNeuroMod uses the Schaefer-1000 atlas. This difference requires us to apply padding to align the feature dimensions when evaluating on CNeuroMod. Furthermore, some concepts in CNeuroMod, such as `telephone`, occur infrequently in BOLD5000, which limits NEURONA's ability to generalize to them. Nonetheless, we find that NEURONA maintains strong performance on queries such as action decoding, suggesting meaningful transfer of motor-related neural representations across datasets.

Table 15: Cross-dataset generalization results, where models are trained on BOLD5000 and tested on CNeuroMod.

|  | Overall | Action | Position | T/F |
|---|---|---|---|---|
| UMBRAE | 0.4494 | 0.0106 | 0.0485 | 0.5036 |
| NEURONA (Ours) | **0.5535** | **0.7237** | **0.5246** | **0.5481** |

## C.2 FMRI RETRIEVAL RESULTS

Here, we detail results from an fMRI retrieval task, where we adapt NEURONA to retrieve the corresponding fMRI given a symbolic query. We use positive queries to ensure that the concept and fMRI are well aligned. Each concept is represented as a one-hot vector over the full concept vocabulary, from which a concept embedding is obtained using a small MLP. We then follow the structure of the symbolic expression by applying an aggregation operation to combine the specified concepts in the query. A parallel MLP encodes the fMRI input into an fMRI embedding, and we train the embedding spaces jointly using a CLIP-based contrastive loss. In this setting, we achieve a test top-1 retrieval accuracy of $0.1325$ and a top-5 accuracy of $0.3012$, substantially outperforming a random-choice baseline (top-1: $0.0120$, top-5: $0.0602$).

Table 16: Network ablations to evaluate the contributions of each network to decoding performance.

| Selected network | Overall | Action | Position | T/F |
|---|---|---|---|---|
| VisCent VisPeri | 0.6750 ± 0.0067 | 0.3577 ± 0.0459 | 0.3765 ± 0.0159 | 0.7330 ± 0.0072 |
| SomMotA/B | 0.6724 ± 0.0069 | 0.3480 ± 0.0317 | 0.3620 ± 0.0131 | 0.7326 ± 0.0079 |
| DorsAttnA/B | 0.6753 ± 0.0073 | 0.3281 ± 0.0786 | 0.3850 ± 0.0282 | 0.7324 ± 0.0076 |
| SalVentAttnA/B | 0.6709 ± 0.0069 | 0.2910 ± 0.0655 | 0.3839 ± 0.0267 | 0.7281 ± 0.0060 |
| LimbicA/B | 0.6669 ± 0.0101 | 0.3379 ± 0.0918 | 0.3512 ± 0.0603 | 0.7282 ± 0.0044 |
| ContA/B/C | 0.6785 ± 0.0038 | 0.3380 ± 0.0808 | 0.4129 ± 0.0173 | 0.7310 ± 0.0035 |
| DefaultA/B/C TempPar | 0.6804 ± 0.0080 | 0.3248 ± 0.0752 | 0.4357 ± 0.0324 | 0.7297 ± 0.0066 |
| All | 0.7102 ± 0.0053 | 0.5965 ± 0.0322 | 0.5378 ± 0.0135 | 0.7425 ± 0.0057 |

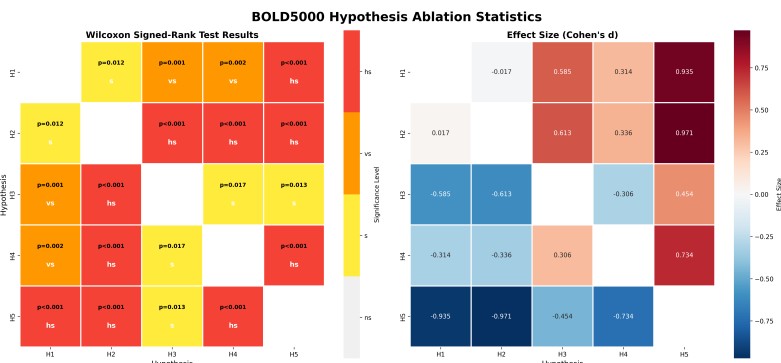

Figure 4: Statistical tests evaluating the effects of the five hypotheses across subjects in BOLD5000.

# D ADDITIONAL ANALYSES

## D.1 NETWORK ABLATIONS

To evaluate how each functional network contributes to decoding performance, we perform a network-level ablation in which NEURONA was trained using only a constrained subset of Yeo-7 networks at a time, isolating the contribution of each network. In Table 16, we see that the top performing networks are the Default Mode, Control, Dorsal Attention, and Visual networks, all of which have been closely linked to visual processing and high-level perceptual representations in prior works Menon (2023); Kucyi et al. (2020); Miyawaki et al. (2008).

## D.2 STATISTICAL TESTS ACROSS HYPOTHESES

We conduct additional statistical analyses for our hypotheses ablations to more rigorously evaluate the effects of the five hypotheses across subjects. Specifically, we use the Wilcoxon signed-rank test to assess statistical significance and Cohen's d to estimate effect size. For each subject, we collect accuracies across all metrics, and perform comparisons between hypotheses using these values. The BOLD5000 and CNeuroMod results are summarized in Figure 4 and Figure 5. Across both datasets, NEURONA's full argument-guided multi-region hypothesis (H5) consistently shows statistically significant improvements over alternative hypotheses.

## D.3 FINE-GRAINED GENERALIZATION ANALYSES

Our generalization experiments on both BOLD5000 (subject-CSI1) and CNeuroMod (subject-01) include argument swapping, predicate transfer, and role systematicity. In Table 17, we report fine-grained systematicity tests across each of these settings. We see that NEURONA shows strong performance on each category, with relatively small drops from all to unseen combinations.

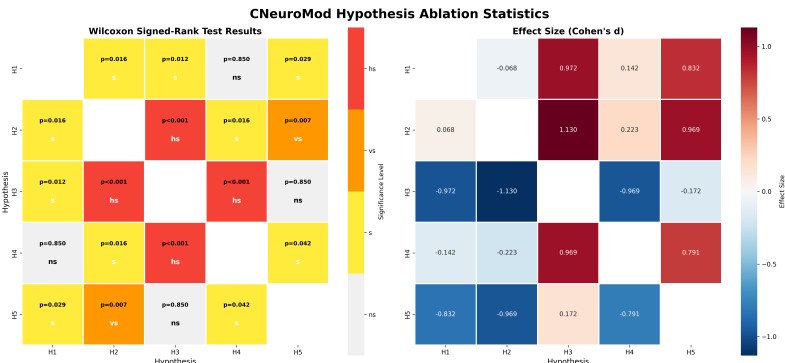

Figure 5: Statistical tests evaluating the effects of the five hypotheses across subjects in CNeuroMod.

Table 17: Fine-grained generalization analyses in BOLD5000-QA and CNeuroMod-QA.

| BOLD5000-QA | Argument swapping | Predicate transfer | Role systematicity | Other | All (T/F) |
|---|---|---|---|---|---|
| Unseen | 0.7451 | 0.75 | 0.7391 | 0.7096 | 0.7184 |
| All | 0.7562 | 0.6667 | 0.8030 | 0.7393 | 0.7407 |
| CNeuroMod-QA | Argument swapping | Predicate transfer | Role systematicity | Other | All (T/F) |
| Unseen | 0.7526 | 0.7333 | 0.7390 | 0.6842 | 0.6991 |
| All | 0.7011 | 0.5088 | 0.7020 | 0.7307 | 0.7250 |

Table 18: Results of predicate argument binding.

| | BOLD5000-QA | CNeuroMod-QA |
|---|---|---|
| Unguided-predicate / subject | -0.0940 | 0.0230 |
| Unguided-predicate / object | 0.0048 | 0.0019 |
| Unguided-predicate / guided-predicate | 0.0336 | 0.1488 |
| Guided-predicate / subject | 0.2020 | 0.1095 |
| Guided-predicate / object | 0.2975 | 0.2552 |

### D.4 PREDICATE ARGUMENT BINDING

We investigate whether unguided predicate representations contain decodable information about their arguments, by computing pairwise correlations between concept groundings across brain regions. For each relational query (e.g., `hold(person, baseball)`), we extract grounding score vectors across functional networks for the subject, object, unguided predicate, and guided predicate. We then compute the correlation matrix between these grounding patterns across all relational queries in both BOLD5000 (4 subjects) and CNeuroMod (3 subjects).

In Table 18, we report the mean correlations, and see minimal correlation between unguided predicate groundings and their subject ($-0.0940$) or object ($0.0048$) arguments in BOLD5000, with similarly low correlations in CNeuroMod ($0.0230$ for subject, $0.0019$ for object). This suggests that when no structural guidance is provided, predicate representations do not bind to specific arguments.

In contrast, guided predicate representations showed substantially higher correlations with both subjects ($0.2020$ in BOLD5000, $0.1095$ in CNeuroMod) and objects ($0.2975$ in BOLD5000, $0.2552$ in CNeuroMod). This increase in correlation suggests that NEURONA's explicit guidance helps decode relations from predicate-argument interactions.

Table 19: Accuracy breakdown between unary and relational concepts in BOLD5000-QA.

| BOLD5000-QA | Overall | Unary | Relation |
|---|---|---|---|
| Yeo-7 | 0.727 | 0.717 | 0.751 |
| Yeo-17 | 0.740 | 0.732 | 0.760 |
| DiFuMo-64 | 0.728 | 0.727 | 0.728 |
| DiFuMo-128 | 0.732 | 0.733 | 0.730 |

Table 20: Accuracy breakdown between unary and relational concepts in CNeuroMod-QA.

| CNeuroMod-QA | Overall | Unary | Relation |
|---|---|---|---|
| Yeo-7 | 0.718 | 0.696 | 0.754 |
| Yeo-17 | 0.725 | 0.707 | 0.754 |
| Schaefer-100 | 0.725 | 0.708 | 0.754 |

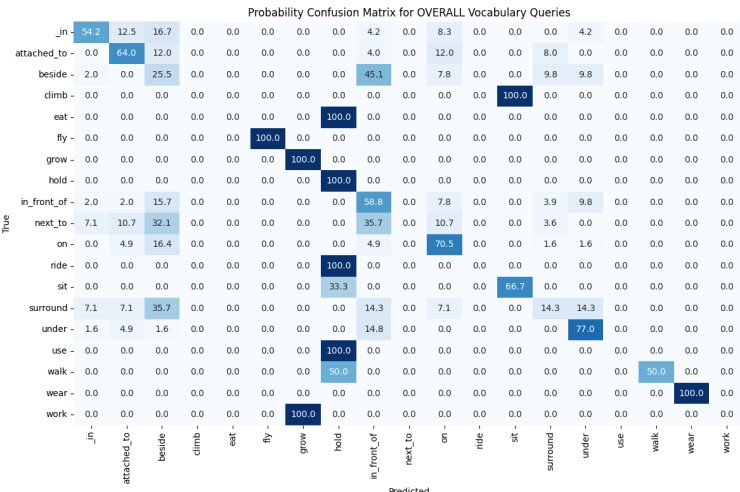

Figure 6: Accuracy confusion matrix across concepts in BOLD5000-QA.

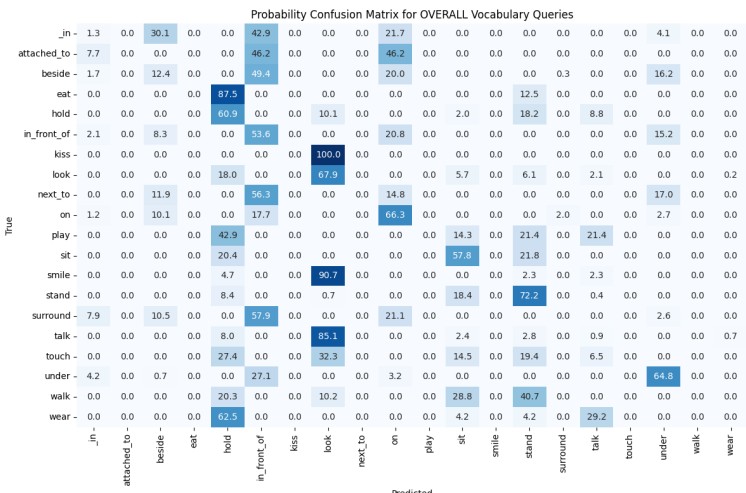

Figure 7: Accuracy confusion matrix across concepts in CNeuroMod-QA

## D.5 DETAILED CONCEPT ACCURACY

In Table 19 and Table 20, we report QA accuracy for unary and relational concepts separately across BOLD5000 and CNeuroMod, to analyze whether query structure affects performance. We see that that performance is generally stable across concept types, and across multiple atlases.

In Figure 6 and Figure 7, we illustrate confusion matrices for both BOLD5000 and CNeuroMod. We see that, as expected, while many concepts are reliably decoded (e.g., visually distinctive actions), some still cause confusion (e.g., spatial relations that are semantically similar).

# E  CONCEPT GROUNDING VISUALIZATIONS

In Figure 8, we present concept grounding examples from the BOLD5000 (Chang et al., 2019) and CNeuroMod (Gifford et al., 2024; Boyle et al., 2023) datasets.

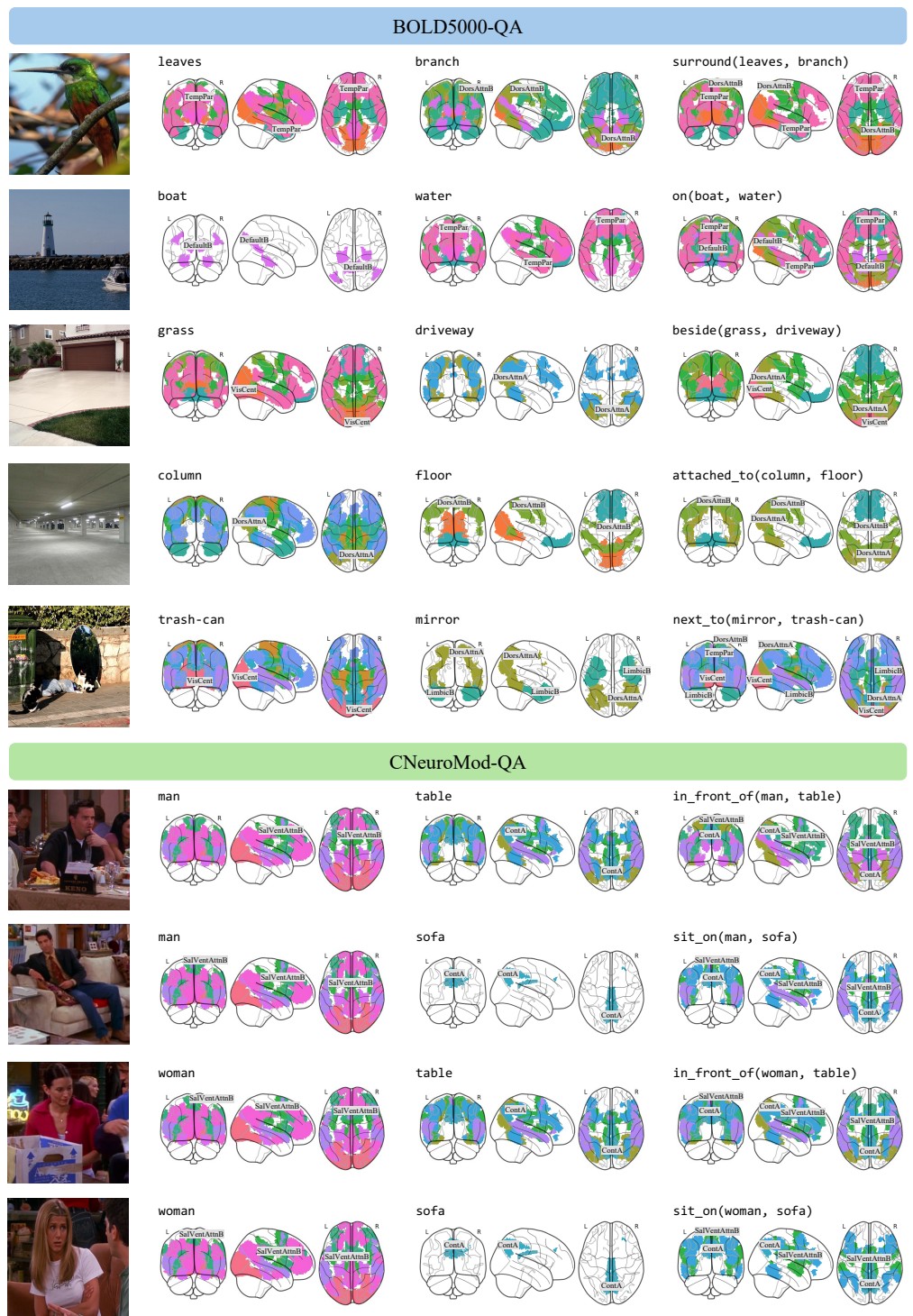

Figure 8: We show examples of learned concept grounding by NEURONA on BOLD5000-QA and CNeuroMod-QA, across subject, object, and predicate concepts.

# F DATASETS

## F.1 LICENSE FOR EXISTING DATASETS

We train NEURONA on the BOLD5000 (Chang et al., 2019) and CNeuroMod (Gifford et al., 2024; Boyle et al., 2023) datasets, which are both licensed under the Creative Commons 0 License. More information can be found on their websites: BOLD5000 and CNeuroMod.

## F.2 FMRI-QA DATASETS

**BOLD5000-QA.** We utilize the BOLD5000 dataset, which has been preprocessed and aligned with image stimuli following WAVE (Wang et al., 2024). The fMRI data has a shape of $[5, 1024]$, representing 5 TRs and 1024 brain regions. We use preprocessed, image-aligned fMRI data provided here. Each TR (repetition time) is 2 seconds, resulting in a chunk duration of 10 seconds ($5 \times 2$s). We account for a hemodynamic lag of 2 TRs. All four subject pairs are included, following the same train-test split as in previous studies.

**CNeuroMod-QA.** We use the CNeuroMod dataset preprocessed by the Algonauts Challenge (Gifford et al., 2024; Boyle et al., 2023). The fMRI data has a TR of $1.49$ seconds and a shape of $[5, 1000]$, representing 5 TRs and 1000 brain regions based on the Schaefer-1000 atlas (Schaefer et al., 2018). This yields a chunk duration of 7.45 seconds ($5 \times 1.49$s), with a hemodynamic lag of 3 TRs. For each chunk, we select the most motion-informative video frame by computing motion energy as the absolute difference between consecutive frames. The chunks are extracted from Friends episodes, with seasons 1–5 used for training and the unseen season 6 reserved for testing. We use $1,000$ fMRI-video chunks per season, resulting in $5,000$ training samples and $1,000$ testing samples.

We provide additional examples from our datasets in Figure 9.

# G EXPERIMENT DETAILS

## G.1 TRAIN SETTINGS

We train and evaluate NEURONA on the specified training and test sets for both BOLD5000 (Chang et al., 2019) and CNeuroMod (Gifford et al., 2024; Boyle et al., 2023). Training is conducted for 100 epochs using the Adam optimizer (Diederik, 2014), with learning rate 0.001 and batch size 32.

## G.2 COMPUTE RESOURCES

Since NEURONA consists of a lightweight convolutional neural network for fMRI feature extraction followed by a linear classifier for concept grounding and execution, its computational requirements are minimal. All experiments are conducted on a single NVIDIA A100 GPU, with training completing in approximately 30 minutes. Data loading is parallelized using 16 CPU workers, and the system uses 64 GB of RAM.

## G.3 BASELINE IMPLEMENTATIONS

We describe the implementation details of the baseline models compared in our study below. In all methods, we treat the fMRI input as a sequence of length 5, with each time step as a token.

**Linear.** We tokenize the input query using the BERT tokenizer (Devlin et al., 2019) and pad all sequences to a fixed length. The tokens are then encoded using a linear layer. We concatenate the fMRI token sequence and the query token sequence, and apply a final linear classification layer to predict the output (either a binary T/F answer or a vocabulary token).

**SDRecon.** We implement SDRecon (Takagi & Nishimoto, 2023) following the official repository[*]. A ridge regression model aligns fMRI features with image embeddings, which are then passed to a VQA-GIT language model (Wang et al., 2022) to generate answers. We set the ridge regularization

---

[*]https://github.com/yu-takagi/StableDiffusionReconstruction

| BOLD5000-QA | |
|---|---|

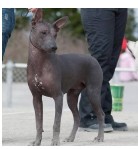 Q: Is there a dog?
A: Yes

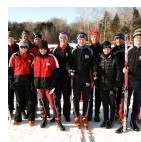 Q: Are there skis on the snow?
A: Yes

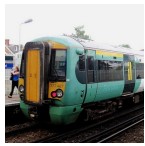 Q: Is there a bus on the road?
A: No

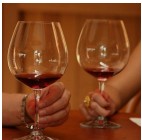 Q: What is the action between the person and the wine glass?
A: Hold

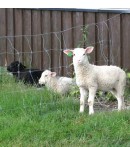 Q: What is the action between the sheep and the grass?
A: Eat

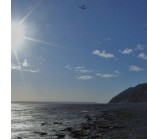 Q: Is there a moon?
A: No

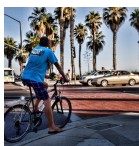 Q: Is there a person riding the motorcycle?
A: No

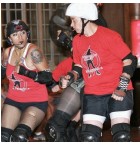 Q: Is there a helmet?
A: Yes

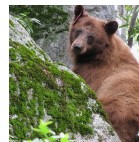 Q: Is there a bear?
A: Yes

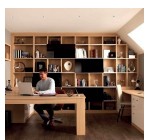 Q: Is there a woman siting on the sofa?
A: No

| CNeuroMod-QA | |
|---|---|

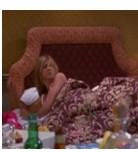 Q: Is there a bed?
A: Yes

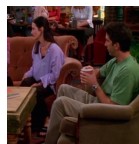 Q: Is there a man in green?
A: Yes

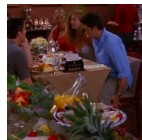 Q: Is there a man sitting on the desk?
A: No

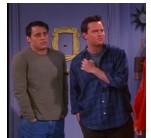 Q: What is the position between the door and the man?
A: Beside

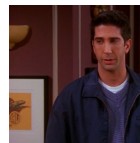 Q: Is there a man looking at the painting?
A: No

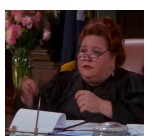 Q: What is the position between the paper and the desk?
A: On

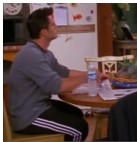 Q: What is the action between the man and the chair?
A: Sit

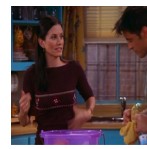 Q: Is there a woman?
A: Yes

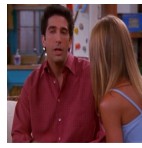 Q: Is there a red shirt?
A: Yes

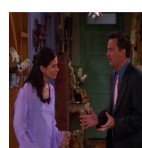 Q: What is the action between the man and the woman?
A: Talk

Figure 9: Examples of queries in BOLD5000-QA and CNeuroMod-QA.

parameter to $\lambda = 20$. A custom parser (described below) is used to map the generated language response to a valid prediction.

**BrainCap.** BrainCap similarly uses a linear encoder to align fMRI features with visual embeddings (Ferrante et al., 2023). The aligned embeddings are passed to a BLIP language model (Li et al., 2022) to generate answers. We apply the same parser as in SDRecon to extract final predictions from the language output. We follow the implementation of the official repository[†].

**UMBRAE.** UMBRAE leverages a transformer-based encoder to map fMRI features to image embeddings (Xia et al., 2024). These embeddings are then passed to LLaVA (Liu et al., 2023) for language-based prediction, followed by response parsing. We follow the implementation of the official repository[‡].

We implement a rule-based parser to convert language model outputs into structured predictions. The parser first cleans the text by removing punctuation, digits, and formatting inconsistencies. It identifies binary answers ("yes" or "no") when the query requires. For other queries, it extracts the first valid word from a predefined vocabulary. If no valid word is found, it defaults to the most common answers of "on" for spatial queries or "hold" otherwise. For all image-grounded baselines, the ground-truth image embeddings are derived from the visual encoder of a vision-language model for BOLD5000. We use the embedding of the first selected video frame as the ground truth for CNeuroMod.

### G.4 ABLATION DETAILS

In this section, we provide the full definitions of the grounding hypotheses introduced in the main paper.

#### G.4.1 ENTITY PROCESSING

To enable concept grounding to neural activity $f \in \mathbb{R}^{N \times T}$, we first map the fine-grained networks $N = 1024$ to $P$ functional networks defined by the given atlas. This results in $P$ network-specific fMRI signals $\{f_1, \ldots, f_P\}$, where each $f_p \in \mathbb{R}^{m_p \times T}$ represents the aggregated signal from $m_p$ fine-grained regions assigned to network $p$. Since the number of regions $m_p$ vary across networks, we apply a linear stitcher to project each $f_p \in \mathbb{R}^{m_p \times T}$ to a fixed-dimensional representation $e_p \in \mathbb{R}^{d \times T}$, where $d = 256$, using network-specific linear projections $W_p \in \mathbb{R}^{m_p \times d}$, such that $e_p = W_p^\top f_p$. This produces a unified set $\mathcal{E}$ of $P$ embeddings $\{e_1, \ldots, e_P\}$, which are then processed by a small 1-D convolutional encoder to form parcellation embeddings. From these base embeddings, we propose hypotheses of candidate regions from which concepts can be grounded.

#### G.4.2 GENERAL GROUNDING FORMULATION

We define the unary grounding score for a concept $c$ as $G_{\text{unary}}(c) \in \mathbb{R}^P$ and the binary (pairwise) grounding score as $G_{\text{binary}}(c) \in \mathbb{R}^{P \times P}$. Since $G_{\text{binary}}(c)$ is defined over parcel pairs, we reduce it to unary space via

$$\tilde{G}_{\text{binary}}(c)_i = \frac{1}{P} \sum_{j=1}^{P} \left[ G_{\text{binary}}(c) \right]_{ij} \in \mathbb{R}^P.$$

We then define a fused grounding score that combines unary and reduced binary evidence:

$$G(c) = G_{\text{unary}}(c) + \tilde{G}_{\text{binary}}(c) \in \mathbb{R}^P. \tag{6}$$

#### G.4.3 HYPOTHESES DEFINITIONS

Let $c_p$, $c_s$, and $c_o$ be the concepts for predicate (e.g., `holding`), subject (e.g., `person`), and object (e.g., `baseball-bat`), respectively.

---

[†] https://github.com/enomodnara/BrainCaptioning
[‡] https://github.com/weihaox/UMBRAE

**H1: Single-region grounding.** Concepts are grounded to a single brain region:

$$
\begin{aligned}
G_{\mathrm{H1}}(c_p) &= G_{\mathrm{unary}}(c_p), \\
G_{\mathrm{H1}}(c_s) &= G_{\mathrm{unary}}(c_s), \\
G_{\mathrm{H1}}(c_o) &= G_{\mathrm{unary}}(c_o).
\end{aligned}
\tag{7}
$$

**H2: Multi-region co-activation.** Concepts are grounded through co-activation across region pairs:

$$
\begin{aligned}
G_{\mathrm{H2}}(c_p) &= \tilde{G}_{\mathrm{binary}}(c_p), \\
G_{\mathrm{H2}}(c_s) &= G_{\mathrm{unary}}(c_s), \\
G_{\mathrm{H2}}(c_o) &= G_{\mathrm{unary}}(c_o).
\end{aligned}
\tag{8}
$$

**H3: Predicate conditioned on subject.** Predicate representations are guided by the activation of the subject region:

$$
\begin{aligned}
G_{\mathrm{H3}}(c_p) &= \tilde{G}_{\mathrm{binary}}(c_p) + G_{\mathrm{unary}}(c_s), \\
G_{\mathrm{H3}}(c_s) &= G_{\mathrm{unary}}(c_s), \\
G_{\mathrm{H3}}(c_o) &= G_{\mathrm{unary}}(c_o).
\end{aligned}
\tag{9}
$$

**H4: Predicate conditioned on object.** Predicate representations are guided by the activation of the object region:

$$
\begin{aligned}
G_{\mathrm{H4}}(c_p) &= \tilde{G}_{\mathrm{binary}}(c_p) + G_{\mathrm{unary}}(c_o), \\
G_{\mathrm{H4}}(c_s) &= G_{\mathrm{unary}}(c_s), \\
G_{\mathrm{H4}}(c_o) &= G_{\mathrm{unary}}(c_o).
\end{aligned}
\tag{10}
$$

**H5: Full argument-guided grounding.** Our proposed method combines multi-region grounding with subject and object guidance. The grounding scores are defined as:

$$
\begin{aligned}
G_{\mathrm{H5}}(c_p) &= G(c_p) + G(c_s) + G(c_o), \\
G_{\mathrm{H5}}(c_s) &= G(c_s), \\
G_{\mathrm{H5}}(c_o) &= G(c_o).
\end{aligned}
\tag{11}
$$

These formulations enable systematic evaluation of how structural priors affect downstream neural decoding accuracy.

# H    ETHICS STATEMENT

This work uses only publicly released fMRI datasets, and focuses on decoding concepts from visual stimuli rather than personal traits or identity, which reduces the risk of individual fingerprinting. However, we acknowledge the inherent risks in building architectures that enable neural decoding from participants. We emphasize that our method is intended solely for scientific analysis, and should not be used for identification or inference of sensitive personal attributes from neural activity.

