# OpenReview forum: "Neuro-Symbolic Decoding of Neural Activity"
_ICLR.cc/2026/Conference — ICLR 2026 Poster_

### Official Review · Reviewer_U1Ce · 2025-10-31

**Soundness:** 3
**Presentation:** 3
**Contribution:** 3
**Rating:** 6
**Confidence:** 4

**Summary:**

The MS introduces a neuro-symbolic fMRI-QA framework that links unary (objects) and relational (predicates) concepts to brain-region embeddings. They show that argument-guided learning of the predicates (i.e., conditioning on subject/object) improves decoding, especially for action/position queries. This is an interesting result indicating the system is role-sensitive and is learning from brain data something like event representations rather than argument-agnostic predicate maps.

Note: I assess the work mainly for its **value in advancing understanding of brain data and brain-research methods**, rather than for engineering improvements in QA performance from fMRI decoders.

**Strengths:**

1. The system learns in a way that understands event semantics, i.e., decoding what is the subject doing to the object.  in fact, while the framing of the work is in terms of learning relational concepts, but I think it's equally fair to say they are learning event semantics, which are driven by relations applied to objects, and the objective is to learn the general semantics.
2. The findings implicitly supports the neurobiological position *against* searching for a general (neurobiological) concept-brain-location in a context independent way, but conditionalizing it on the subject/predicate. This has a major implication for theories that aim to identify brain regions for 'verb meaning' independent of context
3. The neuro-symbolic decomposition is clear and the MS is readable beyond the neuro-symbolic community.

**Weaknesses:**

* There is a main limitation for using the results to inform brain studies, grounded in the use of *Atlases*: using an atlas as the feature basis can (*i*) induce correlations among features because of spatial smoothing and parcellation choices (e.g., 2 brain areas might encode the image in the same way introducing collinearity among features), and (*ii*) allocate variance to parcels that are not informative with respect to the task domain, which limits interpretability and statistical sensitivity.
To my thinking, a better, domain-aligned approach is to start from image-by-voxel data and apply PCA (or other dimensionality reduction; e.g., dictionary learning) prior to training. This would match the feature space to the dataset, perhaps increase precision, and certainly provide more information on distributed brain networks relevant to the concepts. Currently, interpretation is forced to be in the Atlas domain, which is not a state of art basis for brain representations.

**Questions:**

I was wondering if the predicate representation in the unguided setting contains decodable information about the object arguments. If that is the case, it would hint that the predicate embedding/code binds to specific objects.

**Details Of Ethics Concerns:**

The study decodes brain representations form single participants and can, in principle, read out information about novel contents from new participants. This risk however is generic to brain decoding. Here, the datasets are public and not especially sensitive, which mitigates immediate concerns.

Still, a brief ethics statement specific to the architecture would help, to clarify intended use, potential of abuse if used to construct single-person fingerprinting based on the data, and indicating considerations to guard against malicious use, e.g., mapping the features learned per participant to sensitive data.

---

> ### Author Response · Authors · 2025-11-22
>
> Thank you for your constructive feedback, which has helped us greatly improve the quality of our paper. Our revisions based on reviewer comments can be found in blue in our updated manuscript.
>
> **Clarifying use of atlases instead of voxel-level data.**
> We thank the reviewer for the feedback and agree that using fine-grained voxel-level data with PCA as input would indeed be interesting. First, we note that we report experiments across more fine-grained atlases (e.g., DiFuMo-128) in the Appendix, with results including performance on BOLD5000 and CNeuroMod, effect sizes between atlases, and concept grounding consistency across atlases. These results demonstrate robust and modular concept grounding across different levels of spatial granularity, suggesting that our results are not tied to specific parcellation choices.
>
> Second, we highlight that if we treat atlas dimensions as principal components, the fine-grained DiFuMo-1024 atlas explains 70% of voxel-wise variance while maintaining anatomical correspondence [1], which provides a principled and interpretable feature basis for region-wise analyses in neuroscience.
>
> Finally, we also acknowledge that processing millions of whole-brain voxels poses significant computational challenges. Prior visual decoding works [3, 4, 5] typically select only 1,000-15,000 voxels from the visual cortex, rather than using whole-brain signals, partly due to SNR limitations [2]. However, we agree that scaling our method to incorporate whole-brain voxel-level data is a promising next step as the reviewer proposed. We have added these points in the discussion section of the paper.
>
> [1] Dadi, Kamalaker, et al. "Fine-Grain Atlases of Functional Modes for fMRI Analysis." NeuroImage 221 (2020): 117126.
>
> [2] Parrish, Todd B., et al. "Impact of Signal‐to‐Noise on Functional MRI." Magnetic Resonance in Medicine: An Official Journal of the International Society for Magnetic Resonance in Medicine 44.6 (2000): 925-932.
>
> [3] Scotti, Paul, et al. "Reconstructing the Mind's Eye: fMRI-to-Image with Contrastive Learning and Diffusion Priors." Advances in Neural Information Processing Systems 36 (2023): 24705-24728.
>
> [4] Xia, Weihao, et al. "UMBRAE: Unified Multimodal Brain Decoding." European Conference on Computer Vision. Cham: Springer Nature Switzerland, 2024.
>
> [5] Chen, Zijiao, et al. "Seeing Beyond the Brain: Conditional Diffusion Model with Sparse Masked Modeling for Vision Decoding." Proceedings of the IEEE/CVF Conference on Computer Vision and Pattern Recognition. 2023.
>
> **Predicate argument binding.**
> Based on your suggestion, we investigated whether unguided predicate representations contain decodable information about their arguments, by computing pairwise correlations between concept groundings across brain regions. For each relational query (e.g., hold(person, baseball)), we extracted grounding score vectors across functional networks for the subject, object, unguided predicate, and guided predicate. We then computed the correlation matrix between these grounding patterns across all relational queries in both BOLD5000 (4 subjects) and CNeuroMod (3 subjects).
>
> We report the mean correlations in the table below, and see minimal correlation between unguided predicate groundings and their subject (-0.0940) or object (0.0048) arguments in BOLD5000, with similarly low correlations in CNeuroMod (0.0230 for subject, 0.0019 for object). This suggests that when no structural guidance is provided, predicate representations do not bind to specific arguments.
>
> In contrast, guided predicate representations showed substantially higher correlations with both subjects (0.2020 in BOLD5000, 0.1095 in CNeuroMod) and objects (0.2975 in BOLD5000, 0.2552 in CNeuroMod). This increase in correlation indicates that NEURONA's explicit guidance notably helps decode relations from predicate-argument interactions. We include these new results in Appendix A.9.
>
> |                                     | BOLD5000 | CNeuroMod |
> |-------------------------------------|----------|-----------|
> | Unguided-predicate / subject          | -0.0940  | 0.0230    |
> | Unguided-predicate / object           | 0.0048   | 0.0019    |
> | Unguided-predicate / guided-predicate | 0.0336   | 0.1488    |
> | Guided-predicate / subject            | 0.2020   | 0.1095    |
> | Guided-predicate / object             | 0.2975   | 0.2552    |
>
> **Ethics statement.**
> We thank the reviewer for the helpful feedback and have revised our text to include an ethics statement in Appendix E.
>
> Thank you again, and we are happy to help clarify any additional details.

---

### Official Review · Reviewer_FGNG · 2025-10-31

**Soundness:** 3
**Presentation:** 3
**Contribution:** 3
**Rating:** 6
**Confidence:** 3

**Summary:**

This paper presents a neuro-symbolic framework for compositional concept grounding and reasoning using fMRI data. The model aims to decode not only object-level but also relational representations from regional brain activity. The authors evaluate NEURONA on 2 visual fMRI dataset, showing improved decoding and question-answering performance compared to previous fMRI decoding methods (SDRecon, BrainCap, UMBRAE). The work further analyzes the impact of multi-region grounding and relational reasoning through ablation studies.

Overall, the paper is conceptually interesting and methodologically competent, but the empirical depth is relatively limited in statistical validation and subject generalization, and the interpretive link between symbolic composition and neural mechanisms could be made more rigorous. . Additional analyses are highly encouraged.

**Strengths:**

1. The paper is well motivated to tackle the puzzle of interpretable concept grounding in neural data, with an emphasis on compositional structure.
2. The results are consistent across two datasets and multiple concept types.
3. The ablation analyses meaningfully test hypotheses about region-level grounding and compositional generalization.
4. It positions neuro-symbolic modeling as a promising route to study structured neural semantics beyond pixel-level reconstruction.

**Weaknesses:**

1.The results in Tables 1–2 are trained and tested on one subject per dataset, which limits generalizability. Multi-subject or cross-subject validation is essential for neuroscientific conclusions.
2. Grounding is restricted to coarse atlas parcels; voxel-level analyses would strengthen claims about spatial specificity of “modular concepts.”
3. Results in table 1 and 2 are using models trained from one subject in each dataset
4. It remains unclear how much each region or network contributes to concept decoding; e.g., whether “multi-region grounding” corresponds to interpretable brain modules.
5. While means ± SD are reported, formal statistical tests are absent, so robustness to inter-individual variability is uncertain.
6. Without ground-truth region–concept pairs, “consistent” grounding might reflect network-level co-activation rather than explicit compositional structure.
7. Qualitative examples are interesting but could be complemented by quantitative measures (e.g., concept-wise accuracy)
8. Writing is generally clear, though sections 3–5 are dense and occasionally repeat prior claims; discussion could better delineate cognitive vs. computational interpretations.

**Questions:**

1. Have you tried cross-subject evaluations?
2. In the multi-region grounding setup, is there any redundancy or competition between regions (e.g., correlated activations leading to similar grounding weights)?
3. How consistent are the learned concept groundings across subjects — do similar regions emerge for the same concept?
4. Do claims of compositional grounding align with theories of hierarchical organization of the brain?
5. How might this framework extend to other domains (e.g., motor planning)?

---

> ### Author Response · Authors · 2025-11-22
>
> Thank you for your constructive feedback, which has helped us significantly improve the quality of our paper. Our revisions can be found in blue in our updated manuscript.
>
> **Evaluations over all subjects.**
> We thank you for the suggestion. We have added new evaluation over all subjects on both BOLD5000 (4 subjects) and CNeuroMod (3 subjects). For each dataset, we train an individual model for each subject and report the mean ± standard deviation across subjects. We evaluate both (1) in-distribution neural decoding performance and (2) generalization performance. We see that NEURONA significantly outperforms all prior works, similar to trends in Tables 1 and 2. We report these results in Appendix A.4.
>
> | All queries | BOLD5000        |                  |                 |                 | CNeuroMod        |                  |                 |                 |
> |-------------|-----------------|------------------|-----------------|-----------------|------------------|------------------|-----------------|-----------------|
> |             | Overall         | Action           | Position        | T/F             | Overall          | Action           | Position        | T/F             |
> | Linear      | 0.4702 ± 0.0073 | 0.0677 ± 0.0815  | 0.1817 ± 0.0289 | 0.5280 ± 0.0068 | 0.4579 ± 0.0142  | 0.2078 ± 0.0400  | 0.1680 ± 0.0338 | 0.5184 ± 0.0153 |
> | UMBRAE      | 0.4948 ± 0.0098 | 0.2488 ±  0.0572 | 0.2092 ± 0.0401 | 0.5401 ± 0.0097 | 0.4588 ± 0.0425  | 0.1588 ± 0.0102  | 0.1705 ± 0.0220 | 0.5225 ± 0.0248 |
> | SDRecon     | 0.4751 ± 0.0064 | 0.2887 ± 0.0499  | 0.2005 ± 0.0047 | 0.5266 ± 0.0075 | 0.4368 ±  0.0012 | 0.1428 ±  0.0010 | 0.1554 ± 0.0010 | 0.5012 ± 0.0024 |
> | BrainCap    | 0.4842 ± 0.0052 | 0.2412 ± 0.0470  | 0.2005 ± 0.0047 | 0.5383 ± 0.0061 | 0.4422 ± 0.0026  | 0.1245 ± 0.0099  | 0.1497 ± 0.0074 | 0.5110 ± 0.0024 |
> | NEURONA     | **0.7102 ± 0.0053** | **0.5965 ± 0.0322**  | **0.5378 ± 0.0135** | **0.7425 ± 0.0057** | **0.7189 ± 0.0009**  | **0.6422 ± 0.0072**  | **0.5931 ± 0.0084** | **0.7417 ± 0.0005** |
>
> | Generalization | BOLD5000          |                 |                  |                 | CNeuroMod       |                  |                  |                 |
> |----------------|-------------------|-----------------|------------------|-----------------|-----------------|------------------|------------------|-----------------|
> |                | Overall           | Action          | Position         | T/F             | Overall         | Action           | Position         | T/F             |
> | Linear         | 0.4595 ± 0.0093   | 0.0308 ± 0.0312 | 0.1291 ± 0.0545  | 0.5250 ± 0.0059 | 0.4206 ± 0.0129 | 0.0462 ± 0.0654  |  0.1045 ± 0.0667 | 0.4986 ± 0.0052 |
> | UMBRAE         | 0.4886 ± 0.0068   | 0.1214 ± 0.0217 | 0.0914 ± 0.0212  | 0.5392 ± 0.0080 | 0.4487 ± 0.0040 | 0.1107 ± 0.0156  | 0.1386 ± 0.0047  | 0.5247 ± 0.0035 |
> | SDRecon        |  0.4752 ± 0.0064  | 0.2973 ± 0.0361 |  0.2005 ± 0.0047 | 0.5266 ± 0.0075 | 0.4365 ± 0.0014 | 0.1407 ± 0.0008  | 0.1560 ± 0.0012  | 0.5012 ± 0.0019 |
> | BrainCap       | 0.4838 ± 0.0055   | 0.2412 ± 0.0470 | 0.1996 ± 0.0040  | 0.5380 ± 0.0064 | 0.4382 ± 0.0012 | 0.1217 ± 0.0070  | 0.1455 ± 0.0060  | 0.5069 ± 0.0003 |
> | NEURONA        | **0.6812 ± 0.0055**   | **0.4952 ± 0.0682** | **0.4696 ± 0.0190**  | **0.7217 ± 0.0057** | **0.6676 ± 0.0119** |  **0.2916 ± 0.0878** | **0.5377 ± 0.0215**  | **0.7260 ± 0.0072** |
>
> **Cross-subject concept grounding consistency.**
> We thank the reviewer for suggesting evaluation of cross-subject consistency in concept groundings. To evaluate whether concepts are grounded similarly across individuals, we aggregated grounding scores for each concept across all subjects and queries (for concept c appearing N times per subject, we obtained N × S samples where S is the number of subjects). We then computed a cross-subject consistency metric that measures the similarity of the spatial grounding patterns for each concept across individuals. We evaluate NEURONA against a random assignment null model and a multinomial null model that preserves each concept's distribution across regions. Our results show that NEURONA's cross-subject grounding consistency scores, averaged across all concepts, are significantly higher than both null models (p < 0.001), demonstrating that NEURONA's concept groundings converge to more similar sets of regions across participants. We also report these results in Appendix A.4.
>
> | | BOLD5000        | CNeuroMod        |
> |--------------------|-----------------|------------------|
> | Null (random)      | 0.5276 ± 0.0629 |  0.5267 ± 0.0683 |
> | NULL (multinomial) | 0.6045 ± 0.1685 | 0.6481 ± 0.1644  |
> | NEURONA            | 0.7000 ± 0.2514    | 0.7027 ± 0.2165  |

---

> > ### Author Response · Authors · 2025-11-22
> >
> > **Statistical tests across hypotheses.**
> > We conducted additional statistical analyses for Tables 3 and 4 in the main text to more rigorously evaluate the effects of the five hypotheses across subjects. Specifically, we used the Wilcoxon signed-rank test to assess statistical significance and Cohen's d to estimate effect size. For each subject, we collected accuracies across all metrics, and performed comparisons between hypotheses using these values. The BOLD5000 and CNeuroMod results are summarized below. Across both datasets, NEURONA's full argument-guided multi-region hypothesis (H5) consistently shows statistically significant improvements over alternative hypotheses. We report these results in Appendix A.5.
> >
> > | BOLD5000 Wilcoxon signed-rank test | H1 | H2 | H3 | H4 | H5 |
> > |------------|-----|-----|-----|-----|-----|
> > | H1 | - | p=0.012 | p=0.001 | p=0.002 | p<0.001 |
> > | H2 | p=0.012 | - | p<0.001 | p<0.001 | p<0.001 |
> > | H3 | p=0.001 | p<0.001 | - | p=0.017 | p=0.013 |
> > | H4 | p=0.002 | p<0.001 | p=0.017 | - | p<0.001 |
> > | H5 | p<0.001 | p<0.001 | p=0.013 | p<0.001 | - |
> >
> > | BOLD5000 Cohen's d | H1 | H2 | H3 | H4 | H5 |
> > |------------|--------|--------|--------|--------|--------|
> > | H1 | - | -0.017 | 0.585 | 0.314 | 0.935 |
> > | H2 | 0.017 | - | 0.613 | 0.336 | 0.971 |
> > | H3 | -0.585 | -0.613 | - | -0.306 | 0.454 |
> > | H4 | -0.314 | -0.336 | 0.306 | - | 0.734 |
> > | H5 | -0.935 | -0.971 | -0.454 | -0.734 | - |
> >
> >
> > | CNeuroMod Wilcoxon signed-rank test | H1 | H2 | H3 | H4 | H5 |
> > |------------|-----|-----|-----|-----|-----|
> > | H1 | - | p=0.016 | p=0.012 | p=0.850 | p=0.029 |
> > | H2 | p=0.016 | - | p<0.001 | p=0.016 | p=0.007 |
> > | H3 | p=0.012 | p<0.001 | - | p<0.001 | p=0.850 |
> > | H4 | p=0.850 | p=0.016 | p<0.001 | - | p=0.042 |
> > | H5 | p=0.029 | p=0.007 | p=0.850 | p=0.042 | - |
> >
> > | CNeuroMod Cohen's d | H1 | H2 | H3 | H4 | H5 |
> > |------------|--------|--------|--------|--------|--------|
> > | H1 | - | -0.068 | 0.972 | 0.142 | 0.832 |
> > | H2 | 0.068 | - | 1.130 | 0.223 | 0.969 |
> > | H3 | -0.972 | -1.130 | - | -0.969 | -0.172 |
> > | H4 | -0.142 | -0.223 | 0.969 | - | 0.791 |
> > | H5 | -0.832 | -0.969 | 0.172 | -0.791 | - |
> >
> > **Voxel-level analyses.**
> > We thank the reviewer for the feedback and agree that using fine-grained voxel-level data would be interesting. First, we note that we report experiments across more fine-grained atlases (e.g., DiFuMo-128) in the Appendix, with results including performance on BOLD5000 and CNeuroMod, effect sizes between atlases, and concept grounding consistency across atlases. These results demonstrate robust and modular concept grounding across different levels of spatial granularity, suggesting that our results are not tied to specific parcellation choices.
> >
> > Second, we highlight that if we treat atlas dimensions as principal components, the fine-grained DiFuMo-1024 atlas explains 70% of voxel-wise variance while maintaining anatomical correspondence [1], which provides a principled and interpretable feature basis for region-wise analyses in neuroscience.
> >
> > Finally, we also acknowledge that processing millions of whole-brain voxels poses significant computational challenges. Prior visual decoding works [3, 4, 5] typically select only 1,000-15,000 voxels from the visual cortex, rather than using whole-brain signals, partly due to SNR limitations [2]. However, we agree that scaling our method to incorporate whole-brain voxel-level data is a promising next step. We have added these points in the discussion section of the paper.
> >
> > [1] Dadi, Kamalaker, et al. "Fine-Grain Atlases of Functional Modes for fMRI Analysis." NeuroImage 221 (2020): 117126.
> >
> > [2] Parrish, Todd B., et al. "Impact of Signal‐to‐Noise on Functional MRI." Magnetic Resonance in Medicine: An Official Journal of the International Society for Magnetic Resonance in Medicine 44.6 (2000): 925-932.
> >
> > [3] Scotti, Paul, et al. "Reconstructing the Mind's Eye: fMRI-to-Image with Contrastive Learning and Diffusion Priors." Advances in Neural Information Processing Systems 36 (2023): 24705-24728.
> >
> > [4] Xia, Weihao, et al. "UMBRAE: Unified Multimodal Brain Decoding." European Conference on Computer Vision. Cham: Springer Nature Switzerland, 2024.
> >
> > [5] Chen, Zijiao, et al. "Seeing Beyond the Brain: Conditional Diffusion Model with Sparse Masked Modeling for Vision Decoding." Proceedings of the IEEE/CVF Conference on Computer Vision and Pattern Recognition. 2023.

---

> > > ### Author Response · Authors · 2025-11-22
> > >
> > > **Evaluation of network contributions.**
> > > We thank you for your suggestion to evaluate network contributions. To evaluate how each functional network contributes to decoding performance, we performed a network-level ablation in which NEURONA was trained using only a constrained subset of Yeo-7 networks at a time, isolating the contribution of each network. Across these ablations, we see that the top performing networks are the Default Mode, Control, Dorsal Attention, and Visual networks, all of which have been closely linked to visual processing and high-level perceptual representations in prior works [1, 2, 3], supporting the interpretability of NEURONA's multi-region grounding. We add these new results in Appendix A.7.
> > >
> > > | Selected network                   | Overall         | Action           | Position        | T/F             |
> > > |------------------------------------|-----------------|------------------|-----------------|-----------------|
> > > | VisCent VisPeri                    | 0.6750 ± 0.0067 | 0.3577 ± 0.0459  | 0.3765 ± 0.0159 | 0.7330 ± 0.0072 |
> > > | SomMotA SomMotB                    | 0.6724 ± 0.0069 | 0.3480 ± 0.0317  | 0.3620 ± 0.0131 | 0.7326 ± 0.0079 |
> > > | DorsAttnA DorsAttnB                | 0.6753 ± 0.0073 | 0.3281 ± 0.0786  | 0.3850 ± 0.0282 | 0.7324 ± 0.0076 |
> > > | SalVentAttnA SalVentAttnB          | 0.6709 ± 0.0069 | 0.2910 ± 0.0655  | 0.3839 ± 0.0267 | 0.7281 ± 0.0060 |
> > > | LimbicA LimbicB                    | 0.6669 ± 0.0101 | 0.3379 ± 0.0918  | 0.3512 ± 0.0603 | 0.7282 ± 0.0044 |
> > > | ContA ContB ContC                  | 0.6785 ± 0.0038 | 0.3380 ± 0.0808  | 0.4129 ± 0.0173 | 0.7310 ± 0.0035 |
> > > | DefaultA DefaultB DefaultC TempPar | 0.6804 ± 0.0080 |  0.3248 ± 0.0752 | 0.4357 ± 0.0324 | 0.7297 ± 0.0066 |
> > > | All                                | 0.7102 ± 0.0053 | 0.5965 ± 0.0322  | 0.5378 ± 0.0135 | 0.7425 ± 0.0057 |
> > >
> > > [1] Menon, Vinod. "20 Years of the Default Mode Network: A Review and Synthesis." Neuron 111.16 (2023): 2469-2487.
> > >
> > > [2] Kucyi, Aaron, et al. "Electrophysiological Dynamics of Antagonistic Brain Networks Reflect Attentional Fluctuations." Nature Communications 11.1 (2020): 325.
> > >
> > > [3] Miyawaki, Yoichi, et al. "Visual Image Reconstruction from Human Brain Activity Using a Combination of Multiscale Local Image Decoders." Neuron 60.5 (2008): 915-929.
> > >
> > > **Lack of ground truth in concept grounding.**
> > > We agree with the reviewer and highlight in our text that as there is indeed no ground truth region-concept pairs, evaluation of grounding itself is challenging. However, to demonstrate that our concept grounding is meaningful, in Table 5 in the main text, we provide the results of our consistency metric against a null model that randomly assigns each concept to a region subset via uniform sampling. In addition, we now include a stronger null model that uses a multinomial distribution to randomly assign concept groundings across brain networks, preserving the exact total sample count for each concept while distributing samples uniformly across all networks. This ensures that the null model retains the same data structure as the observed data (i.e., same total grounding counts per concept) while randomizing only the network assignments. We run this null model 10 times and report the mean and standard deviation. As shown below, NEURONA similarly significantly outperforms the null model in grounding consistency. We report these results in Appendix A.11.
> > >
> > > |                    | BOLD5000        | CNeuroMod     |
> > > |--------------------|-----------------|---------------|
> > > | Null (multinomial) | 0.6267 ± 0.0067 | 0.6424±0.0057 |
> > > | NEURONA            | 0.8475±0.0186   | 0.8592±0.0084 |

---

> > > > ### Author Response · Authors · 2025-11-22
> > > >
> > > > **Quantitative measures of concept-wise accuracy.**
> > > > We note that we report quantitative unary and relational concept accuracy in Appendix A.8. Based on your suggestion, we have now also added confusion matrices for both BOLD5000 and CNeuroMod. We see that, as expected, while many concepts are reliably decoded (e.g., visually distinctive actions), some still cause confusion (e.g., spatial relations that are semantically similar).
> > > >
> > > > | BOLD5000 True \ Predicted | _in | attached_to | beside | climb | eat | fly | grow | hold | in_front_of | next_to | on | ride | sit | surround | under | use | walk | wear | work |
> > > > |-----------------|-----|-------------|--------|-------|-----|-----|------|------|-------------|---------|-----|------|-----|----------|-------|-----|------|------|------|
> > > > | _in | 54.2 | 12.5 | 16.7 | 0 | 0 | 0 | 0 | 0 | 4.2 | 0 | 8.3 | 0 | 0 | 0 | 4.2 | 0 | 0 | 0 | 0 |
> > > > | attached_to | 0 | 64.0 | 12.0 | 0 | 0 | 0 | 0 | 0 | 4.0 | 0 | 12.0 | 0 | 0 | 8.0 | 0 | 0 | 0 | 0 | 0 |
> > > > | beside | 2.0 | 0 | 25.5 | 0 | 0 | 0 | 0 | 0 | 45.1 | 0 | 7.8 | 0 | 0 | 9.8 | 9.8 | 0 | 0 | 0 | 0 |
> > > > | climb | 0 | 0 | 0 | 0 | 0 | 0 | 0 | 0 | 0 | 0 | 0 | 0 | 100.0 | 0 | 0 | 0 | 0 | 0 | 0 |
> > > > | eat | 0 | 0 | 0 | 0 | 0 | 0 | 0 | 100.0 | 0 | 0 | 0 | 0 | 0 | 0 | 0 | 0 | 0 | 0 | 0 |
> > > > | fly | 0 | 0 | 0 | 0 | 0 | 100.0 | 0 | 0 | 0 | 0 | 0 | 0 | 0 | 0 | 0 | 0 | 0 | 0 | 0 |
> > > > | grow | 0 | 0 | 0 | 0 | 0 | 0 | 100.0 | 0 | 0 | 0 | 0 | 0 | 0 | 0 | 0 | 0 | 0 | 0 | 0 |
> > > > | hold | 0 | 0 | 0 | 0 | 0 | 0 | 0 | 100.0 | 0 | 0 | 0 | 0 | 0 | 0 | 0 | 0 | 0 | 0 | 0 |
> > > > | in_front_of | 2.0 | 2.0 | 15.7 | 0 | 0 | 0 | 0 | 0 | 58.8 | 0 | 7.8 | 0 | 0 | 3.9 | 9.8 | 0 | 0 | 0 | 0 |
> > > > | next_to | 7.1 | 10.7 | 32.1 | 0 | 0 | 0 | 0 | 0 | 35.7 | 0 | 10.7 | 0 | 0 | 3.6 | 0 | 0 | 0 | 0 | 0 |
> > > > | on | 0 | 4.9 | 16.4 | 0 | 0 | 0 | 0 | 0 | 4.9 | 0 | 70.5 | 0 | 0 | 1.6 | 1.6 | 0 | 0 | 0 | 0 |
> > > > | ride | 0 | 0 | 0 | 0 | 0 | 0 | 0 | 100.0 | 0 | 0 | 0 | 0 | 0 | 0 | 0 | 0 | 0 | 0 | 0 |
> > > > | sit | 0 | 0 | 0 | 0 | 0 | 0 | 0 | 33.3 | 0 | 0 | 0 | 0 | 66.7 | 0 | 0 | 0 | 0 | 0 | 0 |
> > > > | surround | 7.1 | 7.1 | 35.7 | 0 | 0 | 0 | 0 | 0 | 14.3 | 0 | 7.1 | 0 | 0 | 14.3 | 14.3 | 0 | 0 | 0 | 0 |
> > > > | under | 1.6 | 4.9 | 1.6 | 0 | 0 | 0 | 0 | 0 | 14.8 | 0 | 0 | 0 | 0 | 0 | 77.0 | 0 | 0 | 0 | 0 |
> > > > | use | 0 | 0 | 0 | 0 | 0 | 0 | 0 | 100.0 | 0 | 0 | 0 | 0 | 0 | 0 | 0 | 0 | 0 | 0 | 0 |
> > > > | walk | 0 | 0 | 0 | 0 | 0 | 0 | 0 | 50.0 | 0 | 0 | 0 | 0 | 0 | 0 | 0 | 0 | 50.0 | 0 | 0 |
> > > > | wear | 0 | 0 | 0 | 0 | 0 | 0 | 0 | 0 | 0 | 0 | 0 | 0 | 0 | 0 | 0 | 0 | 0 | 100.0 | 0 |
> > > > | work | 0 | 0 | 0 | 0 | 0 | 0 | 100.0 | 0 | 0 | 0 | 0 | 0 | 0 | 0 | 0 | 0 | 0 | 0 | 0 |
> > > >
> > > > | CNeuroMod True \ Predicted | _in | attached_to | beside | eat | hold | in_front_of | kiss | look | next_to | on | play | sit | smile | stand | surround | talk | touch | under | walk | wear |
> > > > |-----------------|-----|-------------|--------|-----|------|-------------|------|------|---------|-----|------|-----|-------|-------|----------|------|-------|-------|------|------|
> > > > | _in | 1.3 | 0 | 30.1 | 0 | 0 | 42.9 | 0 | 0 | 0 | 21.7 | 0 | 0 | 0 | 0 | 0 | 0 | 0 | 4.1 | 0 | 0 |
> > > > | attached_to | 7.7 | 0 | 0 | 0 | 0 | 46.2 | 0 | 0 | 0 | 46.2 | 0 | 0 | 0 | 0 | 0 | 0 | 0 | 0 | 0 | 0 |
> > > > | beside | 1.7 | 0 | 12.4 | 0 | 0 | 49.4 | 0 | 0 | 0 | 20.0 | 0 | 0 | 0 | 0 | 0.3 | 0 | 0 | 16.2 | 0 | 0 |
> > > > | eat | 0 | 0 | 0 | 0 | 87.5 | 0 | 0 | 0 | 0 | 0 | 0 | 0 | 0 | 12.5 | 0 | 0 | 0 | 0 | 0 | 0 |
> > > > | hold | 0 | 0 | 0 | 0 | 60.9 | 0 | 0 | 10.1 | 0 | 0 | 0 | 2.0 | 0 | 18.2 | 0 | 8.8 | 0 | 0 | 0 | 0 |
> > > > | in_front_of | 2.1 | 0 | 8.3 | 0 | 0 | 53.6 | 0 | 0 | 0 | 20.8 | 0 | 0 | 0 | 0 | 0 | 0 | 0 | 15.2 | 0 | 0 |
> > > > | kiss | 0 | 0 | 0 | 0 | 0 | 0 | 0 | 100.0 | 0 | 0 | 0 | 0 | 0 | 0 | 0 | 0 | 0 | 0 | 0 | 0 |
> > > > | look | 0 | 0 | 0 | 0 | 18.0 | 0 | 0 | 67.9 | 0 | 0 | 0 | 5.7 | 0 | 6.1 | 0 | 2.1 | 0 | 0 | 0 | 0.2 |
> > > > | next_to | 0 | 0 | 11.9 | 0 | 0 | 56.3 | 0 | 0 | 0 | 14.8 | 0 | 0 | 0 | 0 | 0 | 0 | 0 | 17.0 | 0 | 0 |
> > > > | on | 1.2 | 0 | 10.1 | 0 | 0 | 17.7 | 0 | 0 | 0 | 66.3 | 0 | 0 | 0 | 0 | 2.0 | 0 | 0 | 2.7 | 0 | 0 |
> > > > | play | 0 | 0 | 0 | 0 | 42.9 | 0 | 0 | 0 | 0 | 0 | 0 | 14.3 | 0 | 21.4 | 0 | 21.4 | 0 | 0 | 0 | 0 |
> > > > | sit | 0 | 0 | 0 | 0 | 20.4 | 0 | 0 | 0 | 0 | 0 | 0 | 57.8 | 0 | 21.8 | 0 | 0 | 0 | 0 | 0 | 0 |
> > > > | smile | 0 | 0 | 0 | 0 | 4.7 | 0 | 0 | 90.7 | 0 | 0 | 0 | 0 | 0 | 2.3 | 0 | 2.3 | 0 | 0 | 0 | 0 |
> > > > | stand | 0 | 0 | 0 | 0 | 8.4 | 0 | 0 | 0.7 | 0 | 0 | 0 | 18.4 | 0 | 72.2 | 0 | 0.4 | 0 | 0 | 0 | 0 |
> > > > | surround | 7.9 | 0 | 10.5 | 0 | 0 | 57.9 | 0 | 0 | 0 | 21.1 | 0 | 0 | 0 | 0 | 0 | 0 | 0 | 2.6 | 0 | 0 |
> > > > | talk | 0 | 0 | 0 | 0 | 8.0 | 0 | 0 | 85.1 | 0 | 0 | 0 | 2.4 | 0 | 2.8 | 0 | 0.9 | 0 | 0 | 0 | 0.7 |
> > > > | touch | 0 | 0 | 0 | 0 | 27.4 | 0 | 0 | 32.3 | 0 | 0 | 0 | 14.5 | 0 | 19.4 | 0 | 6.5 | 0 | 0 | 0 | 0 |
> > > > | under | 4.2 | 0 | 0.7 | 0 | 0 | 27.1 | 0 | 0 | 0 | 3.2 | 0 | 0 | 0 | 0 | 0 | 0 | 0 | 64.8 | 0 | 0 |
> > > > | walk | 0 | 0 | 0 | 0 | 20.3 | 0 | 0 | 10.2 | 0 | 0 | 0 | 28.8 | 0 | 40.7 | 0 | 0 | 0 | 0 | 0 | 0 |
> > > > | wear | 0 | 0 | 0 | 0 | 62.5 | 0 | 0 | 0 | 0 | 0 | 0 | 4.2 | 0 | 4.2 | 0 | 29.2 | 0 | 0 | 0 | 0 |

---

> > > > > ### Author Response · Authors · 2025-11-22
> > > > >
> > > > > **Link to hierarchical organization.**
> > > > > While our primary contribution is improving neural decoding accuracy through NEURONA's neuro-symbolic executor, our (computational) concept grounding results do aligns with established theories of cortical hierarchy. Our results show interesting patterns in the decoding process that is consistent with hierarchical processing, with links to theories suggesting that complex representations are built through hierarchical integration [1,2].
> > > > >
> > > > > [1] Badre, David, and Mark D'esposito. "Is the Rostro-Caudal Axis of the Frontal Lobe Hierarchical?" Nature Reviews Neuroscience 10.9 (2009): 659-669.
> > > > >
> > > > > [2] Koechlin, Etienne, Chrystele Ody, and Frédérique Kouneiher. "The Architecture of Cognitive Control in the Human Prefrontal Cortex." Science 302.5648 (2003): 1181-1185.
> > > > >
> > > > > **Extensions to other domains.**
> > > > > Beyond visual decoding, NEURONA's neuro-symbolic approach is a promising future direction for applications in motor and language domains, where priors on compositional structure may be helpful. In motor decoding, complex movements naturally decompose into hierarchical action primitives, for example, a compound action like "move left then rotate clockwise 90 degree" combines two basic motor commands that our framework could explicitly model through symbolic composition. This compositional approach aligns well with recent advances in neuromotor interfaces, where researchers have successfully decoded hand poses and movements from EMG signals [1, 2, 3]. By explicitly representing the hierarchical structure of motor plans, NEURONA could improve generalization to novel movement combinations not seen during training, improving upon a key limitation in current brain-computer interfaces [4]. Similarly, for language decoding, NEURONA's symbolic decomposition naturally maps to linguistic hierarchies. Recent work has demonstrated successful reconstruction of continuous speech [5, 6] and semantic content [7] from neural recordings, but these approaches often rely on end-to-end models. We hypothesize that NEURONA can serve as a more data efficient solution.
> > > > >
> > > > > [1] Kaifosh, Patrick, and Thomas R. Reardon. "A Generic Non-Invasive Neuromotor Interface for Human-Computer Interaction." Nature (2025): 1-10.
> > > > >
> > > > > [2] Salter, Sasha, et al. "emg2pose: A Large and Diverse Benchmark for Surface Electromyographic Hand Pose Estimation." Advances in Neural Information Processing Systems 37 (2024): 55703-55728.
> > > > >
> > > > > [3] emg2qwerty: A Large Dataset with Baselines for Touch Typing using Surface Electromyography
> > > > >
> > > > > [4] Willett, Francis R., et al. "High-performance Brain-to-text Communication via Handwriting." Nature 593.7858 (2021): 249-254.
> > > > >
> > > > > [5] Willett, Francis R., et al. "A High-performance Speech Neuroprosthesis." Nature 620.7976 (2023): 1031-1036.
> > > > >
> > > > > [6] Metzger, Sean L., et al. "A High-performance Neuroprosthesis for Speech Decoding and Avatar Control." Nature 620.7976 (2023): 1037-1046.
> > > > >
> > > > > [7] Tang, Jerry, et al. "Semantic Reconstruction of Continuous Language from Non-invasive Brain Recordings." Nature Neuroscience 26.5 (2023): 858-866.
> > > > >
> > > > > Thank you again, and we are happy to help clarify any additional details.

---

### Official Review · Reviewer_RD4u · 2025-10-31

**Soundness:** 1
**Presentation:** 3
**Contribution:** 2
**Rating:** 4
**Confidence:** 4

**Summary:**

This paper proposes NEURONA, a neuro-symbolic framework for grounding compositional concepts in neural activity using fMRI data. The approach extends Logic-Enhanced Foundation Model (Hsu et al. 2023) to fMRI-based question answering, testing different hypotheses about how concepts are grounded in brain regions through hierarchical predicate-argument dependencies. The authors demonstrate improved decoding accuracy on constructed fMRI-QA datasets derived from BOLD5000 and CNeuroMod, showing particular gains on compositional queries involving actions and spatial relationships.

The version of the paper as of now should be rejected because (1) the central claims about discovering compositional organization in the brain are not well supported by the evidence, (2) the experimental design conflates task performance improvements with evidence of neural representational structure, (3) the "grounding" methodology does not actually demonstrate meaningful concept-to-brain mappings, and (4) key claims about compositionality lack proper validation through decomposability or systematic structural tests. The technical framework represents a potentially valuable contribution to neural decoding, but the interpretive claims about brain organization require much stronger evidence than currently provided.

**Strengths:**

Despite fundamental issues, the paper has notable technical strengths:

1. A well-defined evaluator logic for creating conjunctions that could help in better decoding.
2. Strong empirical performance: 47% relative improvement over baselines is substantial, and the generalization to unseen compositional queries is genuinely impressive evidence that the learned representations support novel combinations.
3. Cross-dataset validation: Testing on both image and video datasets with consistent results strengthens the technical contribution.

**Weaknesses:**

1. The Core Interpretive Problem:

The paper's fundamental flaw is the interpretive leap from 'symbolic structure improves neural decoding' to 'relational meaning in the brain emerges from structured activations guided by hierarchical predicate-argument structure.' The authors consistently conflate improved task performance with evidence of compositional neural mechanisms, but these are different claims requiring different types of evidence.
The 'grounding' results are classification logits indicating which brain regions are statistically useful for predicting concept labels—not neural representations in a meaningfully neuroscientific sense. Classification weights in multivariate neuroimaging do not reflect the underlying neural patterns or information content; they indicate statistical utility for discrimination, not representational structure (see Haufe et al. 2014 NeuroImage). True neural representations involve patterns of activity across voxels that can be decoded, analyzed geometrically, and related to cognitive processes. Furthermore, the authors mischaracterize the prior literature on concept grounding. The foundational studies cited (Huth et al., Mitchell et al., Palatucci et al.) use forward encoding models that predict brain activity from semantic features, allowing legitimate claims about neural encoding. The paper should better distinguish its claims from prior work on concept grounding and be more precise about what type of evidence would support compositional neural organization vs. improved decoding.
The distinction between functional and representational compositionality should be made explicit early in the paper. The current presentation conflates these concepts (e.g., see Lake & Baroni 2018 ICML)

2. Experimental Design Issues:

The datasets retrofit compositional reasoning onto passive viewing data without evidence that participants were performing compositional operations. BOLD5000 participants simply viewed images—they were not actively reasoning about spatial relationships or compositional structure. The "ground truth" comes from automated scene graph parsing, not actual participant cognition. This creates a fundamental disconnect between the experimental setup and claims about compositional cognition in the brain. The danger is that improved decoding may simply reflect visual feature correlations rather than compositional neural processes - without evidence that participants engaged compositional reasoning, there's no basis for compositional interpretations of the neural patterns.
Without participants actively performing compositional reasoning during scanning, there is no basis for claiming that improved decoding of these post-hoc questions reflects compositional neural organization. The authors need to demonstrate that participants were actually engaging compositional processes, not just that symbolic priors help classify brain signals.

The dataset construction follows machine learning conventions (scene graphs → QA tasks) which is appropriate for developing better reasoning systems. However, this methodology is insufficient for claims about brain organization. Without experimental evidence that participants engaged compositional reasoning during scanning, improved decoding of post-hoc compositional questions cannot support claims about compositional neural mechanisms. The authors are applying machine learning validation criteria (task performance) to neuroscientific claims (brain organization) without the experimental controls required for the latter.
Both the issues 1 and 2 could be addressed if voxel/region-wise encoding models (akin to Huth et al.) could be constructed for the concepts from the scene graph parsing and the predicted responses corresponding to the concepts from the model can be treated as grounding/representations and the rest of methodology could be applied (transformation into unary and binary embedding space and executor logic).

3. Missing Evidence for Compositionality:

While the paper implements compositional operations in its executor, it does not demonstrate compositional neural representations. Key missing evidence includes:

3.1 Decomposability: True neuro-symbolic compositional systems (tensor products, vector symbolic architectures) maintain constituent structure that can be systematically recovered through inverse operations. In contrast, the executor combines groundings through non-invertible operations that destroy constituent information. There is no principled way to recover individual concept groundings from the final compositional output, demonstrating that the system implements functional composition without compositional representation. I consider the evaluator logic is a good conjunction recipe but not sure about neuro-symbolic composition though given lack of invertibility. And given that it resembles conjunctive representations, I wasn’t surprised that H5 showed greatest evidence.

3.2 Systematic decomposition: While the authors demonstrate generalization to novel combinations by ensuring disjoint train/test sets, they lack fine-grained analysis of systematic recombination patterns. True compositionality requires systematic relationships between constituents and compounds across all concept types - for example, if the system learns holding(person, bat) and beside(person, tree), it should systematically handle holding(person, tree) and beside(person, bat) with predictable performance relationships. The paper shows aggregate performance on unseen combinations but doesn't analyze whether learned representations support the algebraic manipulation of concepts that characterizes genuine compositional systems. While dataset constraints may limit the extent to which systematic recombination tests are feasible, the authors don't report what subset of such analyses their data could support or acknowledge these limitations when making compositionality claims. Without demonstrating systematic performance patterns across specific recombination types (argument swapping, predicate transfer, role systematicity), the generalization results, while impressive, don't constitute strong evidence for compositional representation.

4. Methodological Concerns:

The consistency metric (how often concepts ground to same regions) is presented as validation, but high consistency could simply reflect stable solutions to the classification task rather than meaningful neural organization (similar to the Haufe et al. critique above). Without independent validation against known neuroscience or comparison to null models with random concept assignments, these results cannot support claims about brain organization.

**Questions:**

1.	Can the authors implement encoding models built from the parsed concepts and use them as grounding/representations?
2.	If strong claims about neurosymbolic representations are to be made: Can fine-grained systematicity tests be done? Is it possible for the evaluator to be inverted to make the output decomposable after binding?
3.	Can better null models be used (with random concept assignments)?
4.	If your answer is no to points 1 and 2, can the focus be changed to methodological contribution and decoding improvements rather than strong claims about brain organization?

---

> ### Author Response · Authors · 2025-11-22
>
> Thank you for your constructive feedback, which has helped us greatly improve the quality of our paper. Our revisions based on reviewer comments can be found in blue in our updated manuscript.
>
> **Clarifications on interpretation.**
> We thank the reviewer for their thoughtful and constructive comments, which have helped us clarify the scope of our work. We have updated our writing in the paper (see revisions in blue), to clarify that our primary contribution is a neuro-symbolic framework for fMRI decoding that achieves significant performance improvements by integrating symbolic reasoning and compositional execution with fMRI grounding techniques. Our results demonstrate functional benefits of incorporating such structural priors into the decoding pipeline, and do not serve as evidence of representational compositionality in neural activity. Overall, we emphasize that NEURONA aims to improve fMRI decoding by leveraging compositional structure, which helps extract information from fMRI data in a more data efficient way. These points have been highlighted throughout our paper.
>
> **Experimental design and preliminary results of encoding models.**
> We thank and agree with the reviewer's point that improved task performance does not by itself directly demonstrate that neural representations are organized compositionally in the cognitive sense. Our revised text clarifies our framing of a neuro-symbolic decoding framework that tests computational hypotheses about guided composition, and achieves significantly improved performance when injecting symbolic structure in the decoding process. We also clarify that in our datasets, participants engaged in passive viewing, limiting the cognitive conclusions that can be drawn. We incorporate these limitations explicitly in our discussion section in the paper.
>
> In addition, we agree that encoding model experiments are indeed interesting with appropriate datasets explicitly designed to probe possible compositional reasoning processes. Although we do not have access to a dataset explicitly designed for this, we conducted two exploratory encoding analyses following the reviewer's suggestion, using positive queries to ensure alignment between concept and fMRI:
>
> 1. fMRI reconstruction from parsed symbolic expressions: Here, we extract concepts from the symbolic expression, and treat each concept as a one-hot vector given the full concept vocabulary. We aggregate concepts according to the structure of the parsed expression and regress from this symbolic embedding to the fMRI signal using an MLP encoder over f \in R^{t,r} where t=5 TRs and r=1024 regions. The resulting test r2 = 0.11 is limited, but demonstrates (very) preliminary ability to encode fMRI responses based on parsed symbolic expressions.
>
> 2. fMRI retrieval from parsed symbolic expressions: As symbolic expressions are not complete descriptions and describe only a subset of semantic content, it is difficult to reconstruct the full fMRI from only partial information. Therefore, we conduct a secondary analysis of retrieving fMRI given the symbolic query. We used the same encoding of concepts as above and an MLP to extract embeddings, as well as a parallel MLP to extract fMRI embeddings from fMRI input. We then used a CLIP-based contrastive loss to train the model. In this setting, we achieve a test top-1 retrieval accuracy of 0.1325 and a top-5 accuracy of 0.3012, substantially outperforming a random-choice baseline (top-1: 0.0120, top-5: 0.0602). We have included these results in Appendix A.12.
>
> While these analyses do not constitute controlled encoding model experiments as used in Huth et al., they represent initial steps toward the reviewer's proposal. To faithfully evaluate brain grounding, where we are able to reconstruct brain regions from neuro-symbolic expressions, we would need new datasets that record fMRI for participants with explicit, factorial manipulations of each component. No available dataset, including BOLD5000 or CNeuroMod, contains such manipulations. Without this, we cannot meaningfully evaluate human reasoning, which we acknowledge as a limitation of our current experimental design.
>
> Hence, we highlight in our paper that our main focus is on significantly improved decoding performance, rather than encoding: we show that injecting structural priors (e.g., argument-guided composition) significantly improves decoding accuracy and generalization performance. We explicitly do not claim that participants performed compositional reasoning or that the brain implements symbolic operators. We appreciate the reviewer's insight and have revised the paper to reflect these distinctions.

---

> ### Author Response · Authors · 2025-11-22
>
> **Evidence of compositionality.**
> The reviewer is correct that our framework does not achieve invertible composition in the sense used in tensor product representations or vector symbolic architectures, and we note that we do not claim to achieve such invertibility in our paper. NEURONA instead uses weighted aggregation operations, and notably, we show that this non-invertible compositional structure serves as an effective inductive bias for neural decoding. Our method focuses on functional compositionality, using compositional structures to improve decoding accuracy and generalization, instead of representational compositionality. NEURONA's grounding results show which regions are useful for prediction, rather than how the brain actually represents these concepts.
>
> In addition, we clarify that our generalization experiments on both BOLD5000 (subject-CSI1) and CNeuroMod (subject-01) do indeed include argument swapping, predicate transfer, and role systematicity as the reviewer mentioned. Based on your suggestion, we now report fine-grained systematicity tests across each of these settings, of which our original generalization results encompasses. We see that NEURONA shows strong performance on each category, with relatively small drops from all to unseen combinations. We include analyses here and have added these results into the Appendix A.10.
>
> | BOLD5000 | Argument swapping | Predicate transfer | Role systematicity | Other  | All (T/F)    |
> |----------|-------------------|--------------------|--------------------|--------|---------|
> | Unseen   | 0.7451            | 0.75               | 0.7391             | 0.7096 | 0.7184  |
> | All      | 0.7562            | 0.6667             | 0.8030             | 0.7393 |  0.7407 |
>
> | CNeuroMod | Argument swapping | Predicate transfer | Role systematicity | Other  | All (T/F)   |
> |-----------|-------------------|--------------------|--------------------|--------|--------|
> | Unseen    | 0.7526            | 0.7333             | 0.7390             | 0.6842 | 0.6991 |
> | All       | 0.7011            | 0.5088             | 0.7020             | 0.7307 | 0.7250 |
>
> **Consistency metric comparison to null models with random concept assignments.**
> We note that in Table 5 in the main text, we provide the results of our consistency metric against a null model that randomly assigns each concept to a region subset via uniform sampling. In addition, following your suggestion, we now include a stronger null model that uses a multinomial distribution to randomly assign concept groundings across brain networks, preserving the exact total sample count for each concept while distributing samples uniformly across all networks. This ensures that the null model retains the same data structure as the observed data (i.e., same total grounding counts per concept) while randomizing only the network assignments. We run this null model 10 times and report the mean and standard deviation. As shown below, NEURONA similarly significantly outperforms the null model in grounding consistency. We include these new results in Appendix A.11.
>
> |                    | BOLD5000        | CNeuroMod     |
> |--------------------|-----------------|---------------|
> | Null (multinomial) | 0.6267 ± 0.0067 | 0.6424±0.0057 |
> | NEURONA            | 0.8475±0.0186   | 0.8592±0.0084 |
>
> **Focus on decoding improvements.**
> We thank the reviewer for the feedback and clarify that the focus of our paper is indeed on NEURONA’s strong decoding improvements, notably in generalization settings. We have made revisions to highlight this more prominently in our paper.
>
> Thank you again, and we are happy to help clarify any additional details.

---

### Official Review · Reviewer_fNVL · 2025-10-31

**Soundness:** 3
**Presentation:** 3
**Contribution:** 3
**Rating:** 6
**Confidence:** 2

**Summary:**

The paper proposes NEURONA, a neuro-symbolic framework for decoding compositional concepts from fMRI by parsing each visual stimulus into a symbolic expression (e.g., predicate–argument relations) and grounding its unary and relational concepts to candidate neural entities derived from cortical parcellations. The authors focused on using QA accuracy as the proxy for concept groundings capabilities. The proposed approach extends LEFT to fMRI-QA so that weak supervision from question-answer pairs can train disentangled concept groundings and an end-to-end executor that composes those groundings to answer queries. The authors conduct experiments to show that the proposed method improves accuracy and generalizes to unseen compositions on BOLD5000-QA and CNeuroMod-QA.

**Strengths:**

I am not familiar with tasks with fMRI, but to the best of my knowledge the framework is new in bringing predicate–argument guidance into an fMRI-QA decoder and testing it via five alternatives within one executor.

The overall evaluation setup looks correct to me: the compositional split is appropriate; train and test use disjoint entity–relation pairs, so it measures true generalization. The ablations are clean and isolate the source of gains: unguided multi-region grounding adds little over a single region, while subject/object guidance and full guidance help most, especially for actions and spatial queries. Results are robust and strong across multiple cortical atlases. The qualitative maps are sensible and aid interpretation.

Overall I think the paper is overall clearly written and the ideas are intuitive to understand despite my unfamiliarities with fMRI-related tasks.

**Weaknesses:**

The primary concern is the novelty about the proposed approach.

The proposed method assembles known pieces: a LEFT-style executor, VLM-derived scene graphs, and standard cortical parcellations. The main new element is predicate–argument guidance and the within-executor hypothesis family that tests it. I acknolwedge that these are thoughtful design choices, but they are not a new model class or theory. That said, the paper still adds value: it gives a clean empirical test of the idea, uses a compositional split that rules out memorization, and offers consistency and cross-atlas checks that others can reuse.

In short, the work is incremental in mechanism yet still useful in evidence and evaluation protocol for the research commnuity.

**Questions:**

Since fillers ultimately come from token sequences, I am wondering have the authors thought about how robust are role actions to multi-token fillers or subword boundary changes?

---

> ### Author Response · Authors · 2025-11-22
>
> Thank you for your constructive feedback, which has helped us greatly improve the quality of our paper. Our revisions based on reviewer comments can be found in blue in our updated manuscript.
>
> **Clarifications on novelty.** We would like to clarify that, although NEURONA builds on the general neuro-symbolic paradigm introduced by LEFT and does not introduce a new broad model class, its executor and grounding mechanism differ substantially from LEFT's design, in ways that are essential for enabling our hypotheses and empirical results. First, LEFT assumes explicit object-centric entities (e.g., segmented objects in 2D/3D scenes), whereas NEURONA must construct neural entities from cortical parcellations and learn their functional roles under different grounding hypotheses. Second, to support our hypotheses tests over uncertain neural entities, NEURONA replaces LEFT's hard logical executor, which relies on min/max aggregation and is appropriate when entities are known, with a softer, role-conditioned aggregation mechanism that guides predicate grounding using subject and object scores. While our overall paradigm extends from prior neuro-symbolic work, the execution semantics and grounding formulation in NEURONA are different and essential to enabling the results presented.
>
> **Robustness and creation of concepts.** We note that in our dataset construction, the VLM is prompted to produce a scene graph for each visual stimulus, and we do not impose any inductive bias on how roles or fillers should be formed. The VLM itself outputs canonicalized concepts (e.g., “baseball bat”), which are treated as modular symbols in NEURONA regardless of how they may be tokenized internally by the model. Indeed, we hypothesize and expect concept binding to become more challenging if concepts were represented at a more fine-grained or subword level, as this would introduce variation into the symbolic space.
>
> Thank you again, and we are happy to help clarify any additional details.

---

### Author Response · Authors · 2025-12-01

We thank the AC and reviewers for their time and efforts. As the discussion section has been halted, we provide a brief summary for the AC. We’ve clarified and addressed questions raised by all reviewers; notably, we expanded our evaluation across subjects and show consistent improvements in decoding accuracy. In particular, we appreciate Reviewer RD4u’s constructive comments on the framing of our contributions. We have clarified throughout the paper that our primary contribution is a neuro-symbolic neural decoding framework: NEURONA is designed to improve fMRI decoding by incorporating compositional structure into the decoding pipeline, and we explicitly do not claim to establish compositional organization in the brain. We now more clearly distinguish functional compositionality in our model from representational compositionality in neural activity, and we emphasize the limitations of existing datasets (e.g., passive viewing) for making claims about neural organization. Corresponding changes have been made in the introduction and discussion sections. We highlight that NEURONA is a state-of-the-art method for neural decoding, showing significant and robust improvements over baselines across datasets, subjects, and generalization settings.

---

### Meta-Review · Area_Chair_vZdC · 2025-12-21

**Summary:**

This paper introduces NEURONA, a neuro-symbolic fMRI decoding framework that injects predicate–argument (subject/object) structure into grounding and execution, yielding large improvements in fMRI question answering and compositional generalization. The contribution is methodological: structured neuro-symbolic decoding improves performance, without claiming compositional representations in the brain.

Strength
* Strong decoding + compositional generalization gains; ablations consistently show predicate–argument (subject/object) guidance is the key improvement.
* Revisions fix the main issue: they now clearly claim functional decoding benefits, not evidence of compositional organization in the brain.
* Added multi-subject results + statistical tests and stronger null models, improving robustness/credibility.

Weakness
* Novelty is incremental (largely a careful integration of known components with role-guided grounding/execution).
* Neuroscience interpretability remains limited: passive-viewing datasets + atlas/parcellation features make “grounding” closer to predictive utility than validated concept-to-brain mapping.

Solid ML-for-neural-decoding contribution with improved framing, would be good for encourage improvements in the area.

**Reviewer Scores:**

n/a

---

### Decision · Program_Chairs · 2026-01-26

Accept (Poster)